# Intrinsic deletion at 10q23.31, including the *PTEN* gene locus, is aggravated upon CRISPR-Cas9–mediated genome engineering in HAP1 cells mimicking cancer profiles

Keyi Geng, Lara G Merino, Raül G Veiga, Christian Sommerauer, Janine Epperlein, Eva K Brinkman, Claudia Kutter

**The CRISPR-Cas9 system is a powerful tool for studying gene functions and holds potential for disease treatment. However, precise genome editing requires thorough assessments to minimize unintended on- and off-target effects. Here, we report an unexpected 283-kb deletion on Chromosome 10 (10q23.31) in chronic myelogenous leukemia-derived HAP1 cells, which are frequently used in CRISPR screens. The deleted region encodes regulatory genes, including *PAPSS2*, *ATAD1*, *KLLN*, and *PTEN*. We found that this deletion was not a direct consequence of CRISPR-Cas9 off-targeting but rather occurred frequently during the generation of CRISPR-Cas9–modified cells. The deletion was associated with global changes in histone acetylation and gene expression, affecting fundamental cellular processes such as cell cycle and DNA replication. We detected this deletion in cancer patient genomes. As in HAP1 cells, the deletion contributed to similar gene expression patterns among cancer patients despite interindividual differences. Our findings suggest that the unintended deletion of 10q23.31 can confound CRISPR-Cas9 studies and underscore the importance to assess unintended genomic changes in CRISPR-Cas9–modified cells, which could impact cancer research.**

## Introduction

The CRISPR-Cas system is a widely used genome engineering technology because of its simple programmability, versatile scalability, and targeting efficiency (Wang & Doudna, 2023). Although researchers are rapidly developing CRISPR-Cas9 tools, the biggest challenge remains to overcome undesired on- and off-targeting outcomes. Previous studies have reported unintended genomic alterations, such as, integration of functional target-derived sequences (Geng et al, 2022), large deletion at the double-strand break (DSB) site (Kosicki et al, 2018), chromothripsis (Leibowitz et al,

2021), segmental chromosomal losses (Zuccaro et al, 2020) or translocations (Brunet & Jasin, 2018) that often co-occur (Geng et al, 2022). Most of these genomic rearrangements remain undetectable by conventional validation methods, underscoring the need to thoroughly assess CRISPR-Cas9–modified cells or organisms by more advanced tools.

The complexity of these genomic outcomes is linked to the experimental model system. For example, dysfunctional repair mechanisms in certain cell lines can influence the cellular preference for employing a specific repair pathway, which can result in different editing outcomes (Meyenberg et al, 2021). Furthermore, in polyploid cells, several on-target alterations can occur on various alleles, resulting in diverse phenotypes and biological consequences across cells (Geng et al, 2022). Therefore, the choice of the cellular model system is crucial for investigating biological processes or diseases.

The chronic myeloid leukemia-derived HAP1 cell line is frequently used in genetic studies and large-scale CRISPR screens because its near-haploid genotype increases the probabilities of acquiring modified cells with homozygous genotypes. Homozygous editing is often preferred when biological functions of the CRISPR-Cas9–modified target are assessed (Essletzbichler et al, 2014; Sun et al, 2020). In contrast, a targeted region or gene may only be successfully modified on one allele in diploid or polyploid cells. Thus, the remaining functional copy on the other allele(s) can mask the effect of the CRISPR-Cas9–modified copy, impeding the identification of measurable phenotypic changes and preventing the correct assessment of gene functionality. Given the widespread use of HAP1 in CRISPR-Cas9 applications, it is crucial to eliminate any confounding genetic variances that could jeopardize any experimental conclusions.

In this work, we report the inconsistent occurrence of a large and unexpected genomic deletion at 10q23.31 in HAP1 cells. This deletion was accompanied by widespread changes on the chromatin level and in gene expression. Instead of a commonly reported gRNA-dependent off-targeting event, we found that the generation

---

Department of Microbiology, Tumor, and Cell Biology, Science for Life Laboratory, Karolinska Institute, Solna, Sweden

Correspondence: claudia.kutter@ki.se

 

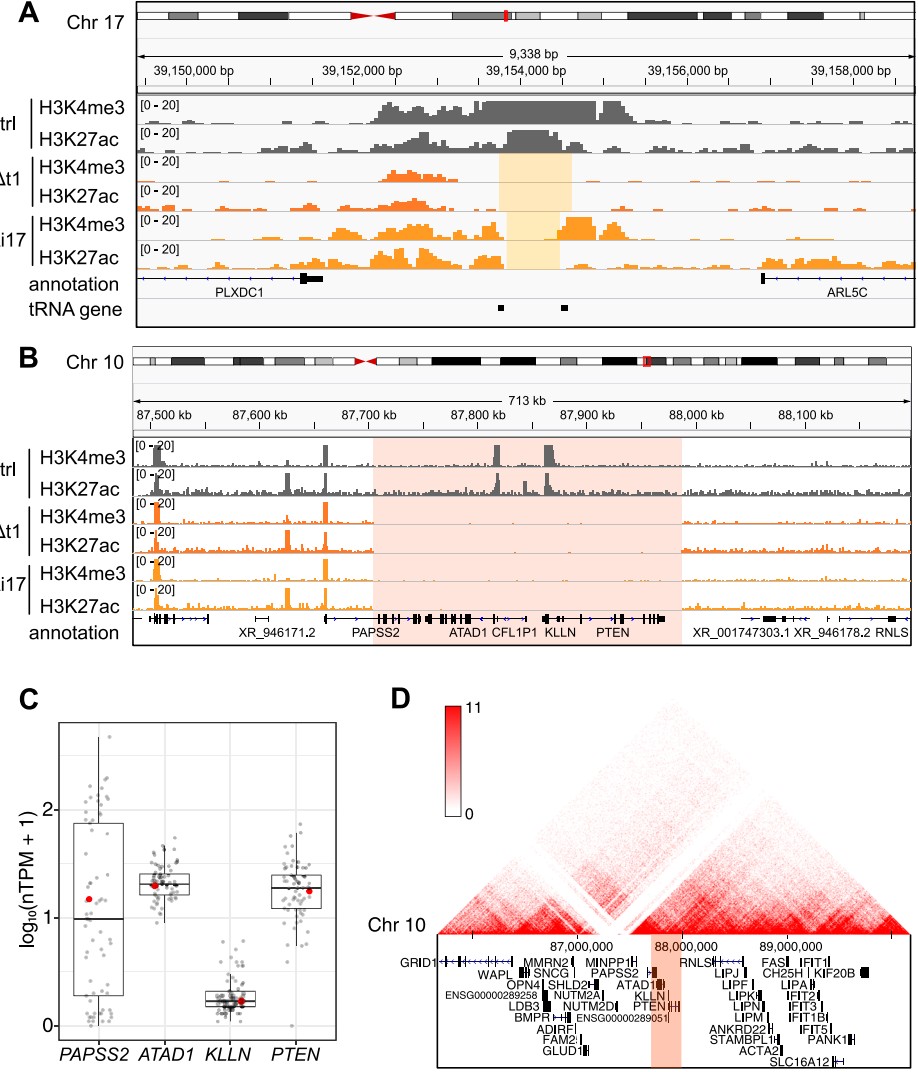

**Figure 1. The loss of the *PAPSS2-PTEN* locus occurred in various CRISPR-Cas9–modified genotypes in HAP1 cells.**
**(A, B)** The hg38 Genome Browser view shows normalized H3K4me3 and H3K27ac ChIP-seq reads in control (ctrl, dark grey), Δt1 (orange), and Δi17 (yellow) HAP1 cell clones at (A) the targeted gene locus on Chr 17 (beige box) and (B) the *PAPSS2-PTEN* locus on Chr 10 (red box). **(C)** Box plot shows the normalized gene expression values (in normalized TPM) of the four protein-coding genes within the deleted region on Chr 10. Each dot represents gene expression values across different cell lines (grey) including HAP1 (red). Normalized TPM values were obtained from the Human Protein Atlas (v. 21.0). Median (horizontal line), interquartile range (box), lower and upper quartiles (whiskers) are shown. **(D)** The genome browser presents a 3D-contact map of Hi-C data at the *PAPSS2-PTEN* locus in HAP1 cells (genomic contact frequency, red: high, white: low). The deleted region is highlighted (red box).

of CRISPR-Cas9 edited cells and/or exposure to cellular stressors greatly increases the occurrence of the deletion. We found the 10q23.31 deletion not only in HAP1 cells but also detected its frequent occurrence across several cancer types in patients. Our findings highlight the importance of considering collateral deletions when assessing mechanistic functions of genes or regulatory regions in cells commonly used for basic research or newly isolated from patients for personalized medicine.

## Results

### CRISPR-Cas9–modified HAP1 cells contained an unexpected 10q23.31 deletion

To study two proximal transfer RNA (tRNA) genes on Chromosome (Chr) 17 (17q12), we previously used the dual gRNA system to generate two variants of CRISPR-Cas9–modified and single cell–derived HAP1 clones in which the genomic region with (Δt clones) or

in between (Δi clones) two tRNA genes was removed (Fig 1A) (Geng et al, 2022). We profiled genomic occupancy of histone (H) 3 lysine (K) 4 trimethylation (H3K4me3) and K27 acetylation (H3K27ac) by chromatin immunoprecipitation-sequencing (ChIP-seq). As intended, H3K4me3 and H3K27ac ChIP-seq enrichment and background signals were absent in between the targeted regions on Chr 17 in the Δt1 and Δi17 clones but not in the CRISPR-Cas9–unmodified control (ctrl) clone confirming the successful deletion of the corresponding genomic regions (Figs 1A and S1A). Unexpectedly, ChIP-seq enrichment or background signals were also undetectable on Chr 10 (10q23.31) in the Δt1 and Δi17 clones but not in the unmodified cell clone (Figs 1B and S1B–F). Because the gRNA sequences (gRNAs) used for generating the Δt and Δi cell clones were different, we concluded that the 10q23.31 deletion occurred in a gRNA sequence-independent manner (Table S1).

To further assess the frequency of the 10q23.31 deletion, we inspected our other CRISPR-Cas9–modified HAP1 cell clones by PCR using primers annealing within the deleted genomic region (Table S1). The absence of a PCR amplicon in 61% (14/23) Δt and 44% (4/9)

Δi clones suggested frequent losses of the 10q23.31 region in our CRISPR-Cas9–modified cell clones (Fig S2A). By comparing the frequencies of the 10q23.31 region in the HAP1 unmodified cell population with HAP1 CRISPR-Cas9–modified single cell-derived clones with the 10q23.31 region, we detected a lower frequency of the 10q23.31 locus within the unmodified HAP1 cell population. This indicates that a fraction of HAP1 cells within the larger population may already possesses the deletion (Fig S2B).

The deleted 10q23.31 locus of about 283 kb encompassed four protein-coding genes (*PAPSS2*, *ATAD1*, *KLLN*, and *PTEN*) and one pseudogene (*CFL1P1*). The four protein-coding genes were widely expressed in 69 cell lines of different tissue origins (Fig 1C) and exert diverse molecular functions. As previously reported, the PAPSS2 enzyme controls the sulphate activation pathway (Kurima et al, 1999; Xu et al, 2002), the transmembrane helix translocase ATAD1 removes mislocalized proteins from the mitochondrial outer membrane (Chen et al, 2014), KLLN regulates cell cycle and apoptosis (Cho & Liang, 2008), and the tumor suppressor PTEN acts as a protein phosphatase (Li et al, 1997; Myers et al, 1997). Furthermore, Hi-C and ChIA-PET data obtained from HAP1 and other cell types revealed multiple long-range interactions between the 10q23.31 and other genomic regions indicating additional roles in three-dimensional (3D) gene regulation (Figs 1D and S2C). Lastly, according to our H3K4me3 and H3K27ac profiling, the deletion ranged from the first intron of the *PAPSS2* gene to a genomic site downstream of the *PTEN* gene. We therefore referred to this genomic deletion as Δ*PAPSS2-PTEN*.

In sum, the *PAPSS2-PTEN* gene locus was frequently deleted in CRISPR-Cas9modified cell clones. Given that this region is important in 3D genome organization and encodes four protein-coding genes with important molecular functions, its unintended deletion may dominate over the expected gRNA-mediated CRISPR-Cas9 genomic deletion and could lead to biological misinterpretations.

### Δ*PAPSS2-PTEN* cells showed abnormal transcript signatures

Because we identified the deletion of the *PAPSS2-PTEN* locus through chromatin profiling, we next examined the impact of the deletion on the transcriptome. In alignment with our ChIP-seq results, our RNA-seq data confirmed the complete loss of gene expression at the *PAPSS2-PTEN* locus in the Δt1 clone when compared with the control cell clone (Fig 2A, top four tracks). Further inspection revealed reads mapping to the first exon of *PAPSS2*, confirming that the genomic region including the promoter, transcriptional start site (TSS), and first exon of *PAPSS2* remained intact (Figs 1B and 2B, top two tracks). Furthermore, we found reads mapping to the positive strand downstream of the *PTEN* gene body in the Δt1 clone, but not in the control cell clone (Fig 2C, top two tracks). This transcript signature was likely caused by polymerase II (Pol II) readthrough from the altered *PAPSS2* gene locus. This is because Pol II can still be recruited to the *PAPSS2* promoter, leading to the initiation of aberrant transcript formation in Δ*PAPSS2-PTEN* cells. However, the *PAPSS2* Pol II termination signal and major parts of the *PAPSS2* gene body were lost together with *ATAD1*, *KLLN*, and *PTEN*.

To further inspect our newly identified genome and transcript signature at the Δ*PAPSS2-PTEN* region, we searched for published RNA-seq data of CRISPR-Cas9-modified HAP1 cells. We retrieved RNA-seq datasets from three independent studies in which various genes were modified, affecting the genes *METAP1* (ΔM) (dataset 1), *C12orf49* (ΔC) and *SREBF2* (ΔS) (dataset 2) as well as *SMARCC1* (ΔSM1) and *SMARCC2* (ΔSM2) (dataset 3) (Table S2). No direct network interactions were found between those CRISPR-Cas9–modified genes and the Δ*PAPSS2-PTEN*–encoded genes (Fig S3A). Similar to our Δt1 clone, we found that the *PAPSS2-PTEN* locus was also deleted in the ΔM, ΔS, and ΔSM1 but not in the ΔC and ΔSM2 clones (Fig 2A and B).

To identify common gene expression signatures associated with the deletion of the *PAPSS2-PTEN* locus, we combined ours and the other three datasets. We performed a principal component analysis (PCA). As commonly observed, PC1 and PC2 separated the samples by dataset likely because of technical biases (Leek et al, 2010), such as, sample and library preparation or sequencing (Fig S3B). However, the subsequent PCs (PC3–PC5) showed that samples with the *PAPSS2-PTEN* locus deletion (Δ*PAPSS2-PTEN* group) clustered together despite their differences in the CRISPR-Cas9–modified genotypes (Fig S3C and D). We corrected for batch biases to minimize the differences between the datasets that were introduced during sample and library preparation. After the batch correction, PC1 and PC2 classified the samples either in the *PAPSS2-PTEN* deletion-positive (Δ*PAPSS2-PTEN*) or in the -negative (with *PAPSS2-PTEN*) group (Fig 2D). We next examined the top 20 genes contributing to PC1 and PC2 and hence to the separation of the two groups. *ATAD1*, *PTEN*, and *PAPSS2* were among the top genes that contributed to the strongest separation of the two groups (Fig 2D). None of the top genes included the originally intended gene knockouts (Fig 2D). Besides *ATAD1*, *PTEN*, and *PAPSS2*, the other top 20 genes were located on different chromosomes (Tables S3 and S4).

Thus, our inspection of transcriptome data from various CRISPR-Cas9–modified HAP1 cell clones confirmed a gRNA sequence-independent deletion at the *PAPSS2-PTEN* locus. Importantly, the unexpected *PAPSS2-PTEN* deletion resulted in similar gene expression changes dominating over the intended gene modification, which could bias the assessment of gene functionality.

### Gene expression changes in Δ*PAPSS2-PTEN* HAP1 cells affected fundamental processes including cell cycle and DNA replication

To obtain a comprehensive understanding of the transcriptional changes in Δ*PAPSS2-PTEN* HAP1 cells, we performed a differential gene expression analysis. We found a total of 2,918 differentially expressed (DE) genes corresponding to 1,489 down- and 1,429 up-regulated genes (Fig 3A) located on different chromosomes with no apparent positional clustering (Fig 3B). A few DE genes were located on Chr 10 but resided in topologically associating domains (TADs) other than the TAD encompassing the *PAPSS2-PTEN* locus. For example, *SNCG* is located in linear distance closest (700 kb) and *MALRD1* furthest (6,800 kb) to the *PAPSS2-PTEN* locus (Figs 1D and 3C). These results suggested that the complex transcriptional changes observed in Δ*PAPSS2-PTEN* cells were not caused by local effects of the deletion on nearby gene expression.

To assess whether specific regulatory processes were changed upon the deletion of the *PAPSS2-PTEN* locus, we

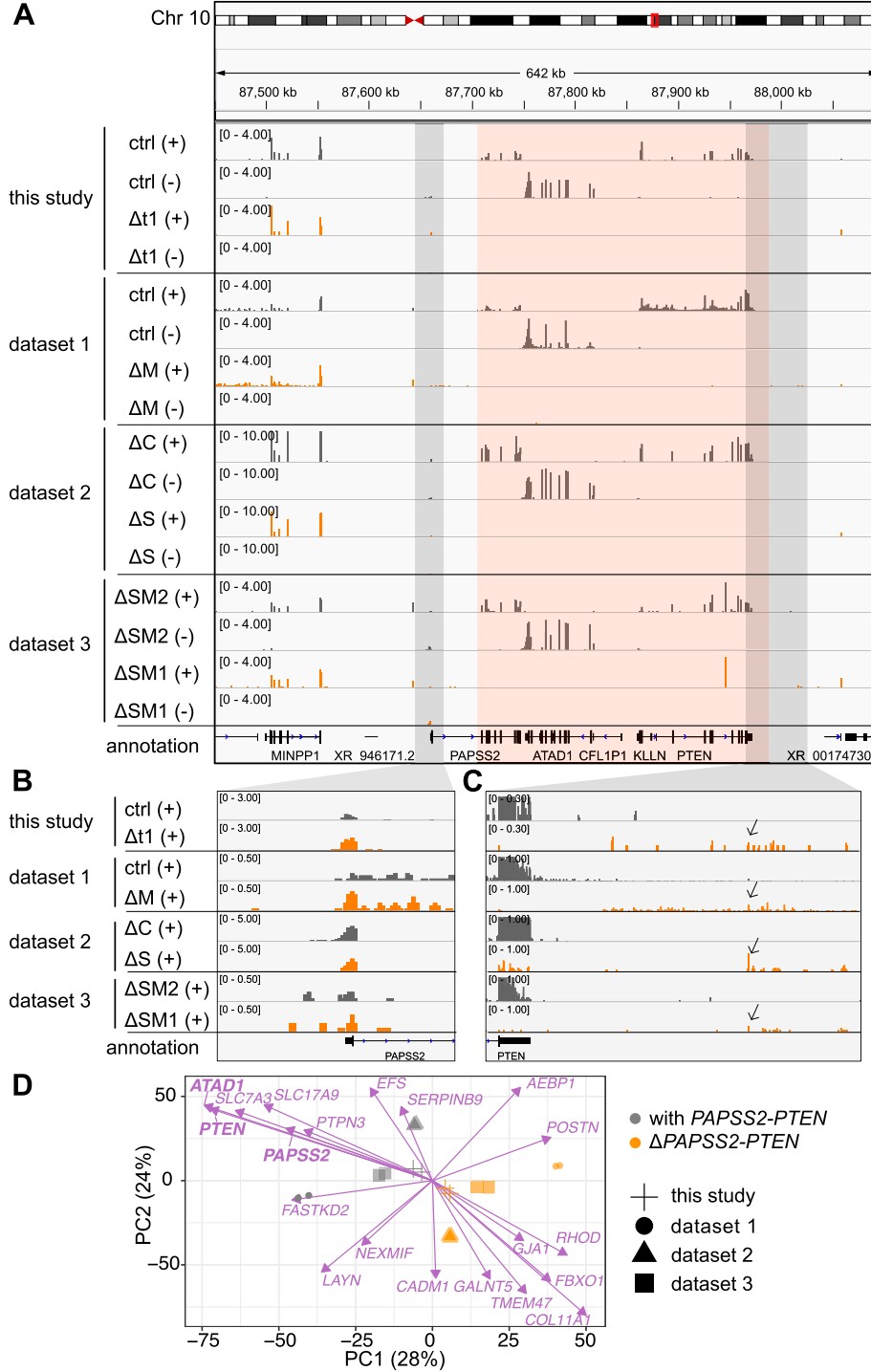

**Figure 2. The unintended loss of the *PAPSS2-PTEN* locus resulted in similar transcriptional changes despite genotypical differences in CRISPR-Cas9–modified HAP1 cell clones.**
**(A, B, C)** The hg38 genome browser view shows normalized RNA-seq coverage tracks for the plus (+) and minus (−) strand over the *PAPSS2-PTEN* locus (highlight in red box) for CRISPR-Cas9–modified (dark grey) and control (ctrl) HAP1 cell clones (orange) generated in this and other studies (dataset 1–3). **(B, C)** The (B) upstream and (C) downstream regions of the *PAPSS2-PTEN* locus (indicated as light grey boxes in Fig 2A) are magnified and visualized for the plus strand. Pol II read-through signals are indicated (arrows). **(D)** Factorial map of the principal component analysis after batch effect correction is shown for the top 20 genes (purple arrows) separating the HAP1 cell clones that contain (with, grey circle) or lost (Δ, orange circle) the *PAPSS2-PTEN* locus in our and other datasets (geometric shapes) in PC1 and PC2. The proportion of variance explained by each PC is indicated in parenthesis.

grouped DE genes according to gene ontology (GO) terms (Fig 3D, Tables S5, S6, S7, and S8). The down-regulated genes were significantly enriched for biological processes, such as DNA replication, cell cycle, and DSB repair and molecular functions, including DNA and histone binding. In contrast, the up-regulated genes significantly controlled biological processes and molecular functions linked to GO terms such as development and catalytic activities. Accordingly, our Kyoto Encyclopedia of Genes and Genomes pathway enrichment analysis confirmed the identified GO terms (Fig 3E, Tables S9 and S10). Many DE genes, such as replication factor C (*RFC1*, *RFC3*, and *RFC3*), the *MCM* gene family and genes forming DNA polymerase subunits, controlled crucial cell processes, and contributed to the five most enriched pathways (Fig 3F).

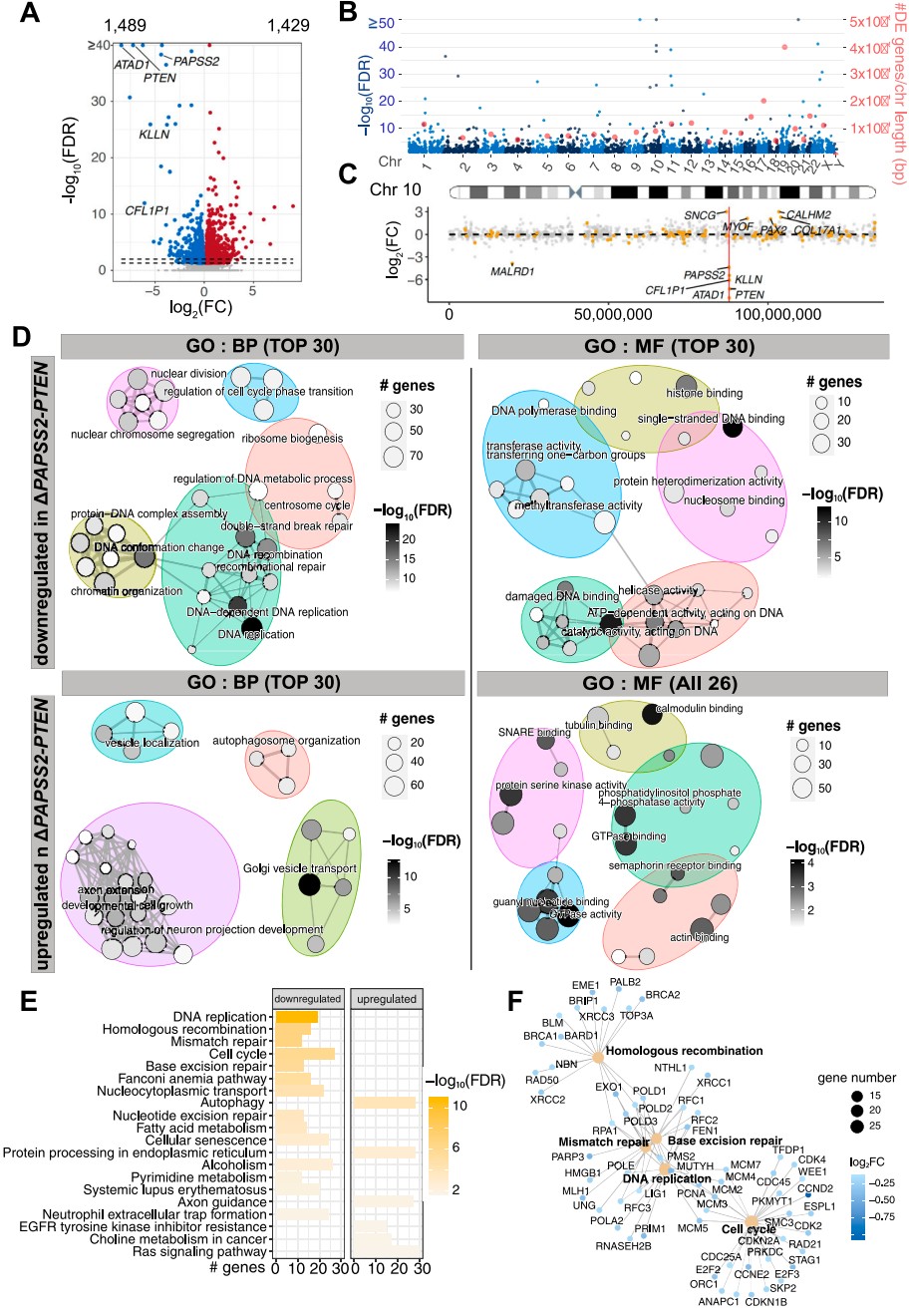

**Figure 3. The *PAPSS2-PTEN* locus deletion was associated with confounding transcriptomic alterations impacting molecular processes.**
**(A)** Volcano plot separates DE genes when comparing transcriptomes of HAP1 cell clones with and without the *PAPSS2-PTEN* deletion. Each dot shows non-DE genes (grey) or DE genes (red: $\log_2$[fold-change, FC] > 0 and blue: $\log_2$[FC] < 0, FDR-adjusted *P*-values, FDR ≤ 0.05). The numbers of up- and down-regulated genes are labelled above the plot and genes located in the deleted 10q23.31 region are highlighted. The two dashed lines indicate FDR-adjusted *P*-values (FDR = 0.05 and FDR = 0.01). **(B)** Manhattan plot shows the chromosomal distribution (light and dark blue dots) and frequency of the DE genes normalized by chromosome length (red dots). **(C)** Dot plot shows the distribution on Chr 10 (x-axis) and fold change (y-axis) of non-DE (grey) and DE genes (orange). The red line highlights the location of the deleted *PAPSS2-PTEN* locus. DE genes with |$\log_2$(FC)| > 2 are labelled. **(D)** Enrichment maps illustrate the up to 30 or 26 most significantly enriched biological process (BP) and molecular function (MF) GO terms for down- and up-regulated DE genes. Each node represents one enriched GO term. The sizes of the nodes are determined by the numbers of DE genes contributing to the enriched GO term. The nodes are colored based on FDR-adjusted *P*-values. **(E)** Bar plot shows the 20 most enriched Kyoto Encyclopedia of Genes and Genomes pathways for down- and up-regulated genes (ranked by FDR-adjusted *P*-values). **(F)** The gene concept network displays genes contributing to the top five enriched Kyoto Encyclopedia of Genes and Genomes pathways (ranked by FDR-adjusted *P*-values). The node diameter determines the number of genes, colored according to the fold change.

Because our batch effect correction could have introduced biases, we performed the DE analysis for each dataset separately by comparing the groups with and without the *PAPSS2-PTEN* locus. Most (88%, 2,570 of 2,918) of the DE genes was commonly deregulated between the pooled batch effect-corrected datasets and in each separately analyzed dataset (Fig S3E). Importantly, DE genes identified in at least three of the four separately analyzed datasets were significantly enriched in biological processes, comprising cell cycle, DNA replication, DNA repair, and molecular functions, including single-strand DNA binding, which was in accordance with the results obtained after batch effect correction (Fig S3F).

In conclusion, transcriptome signatures in CRISPR-Cas9–modified cell clones carrying the genomic deletion of the *PAPSS2-PTEN* locus were profoundly altered.

## Gene expression and H3K27ac changes were linked in *ΔPAPSS2-PTEN* HAP1 cells

Because our GO enrichment analysis suggested changes in chromatin organization and modification, we expected alterations in genome accessibility in the *ΔPAPSS2-PTEN* cell clones. We therefore quantified genomic occurrences of H3K27ac, which demarcates

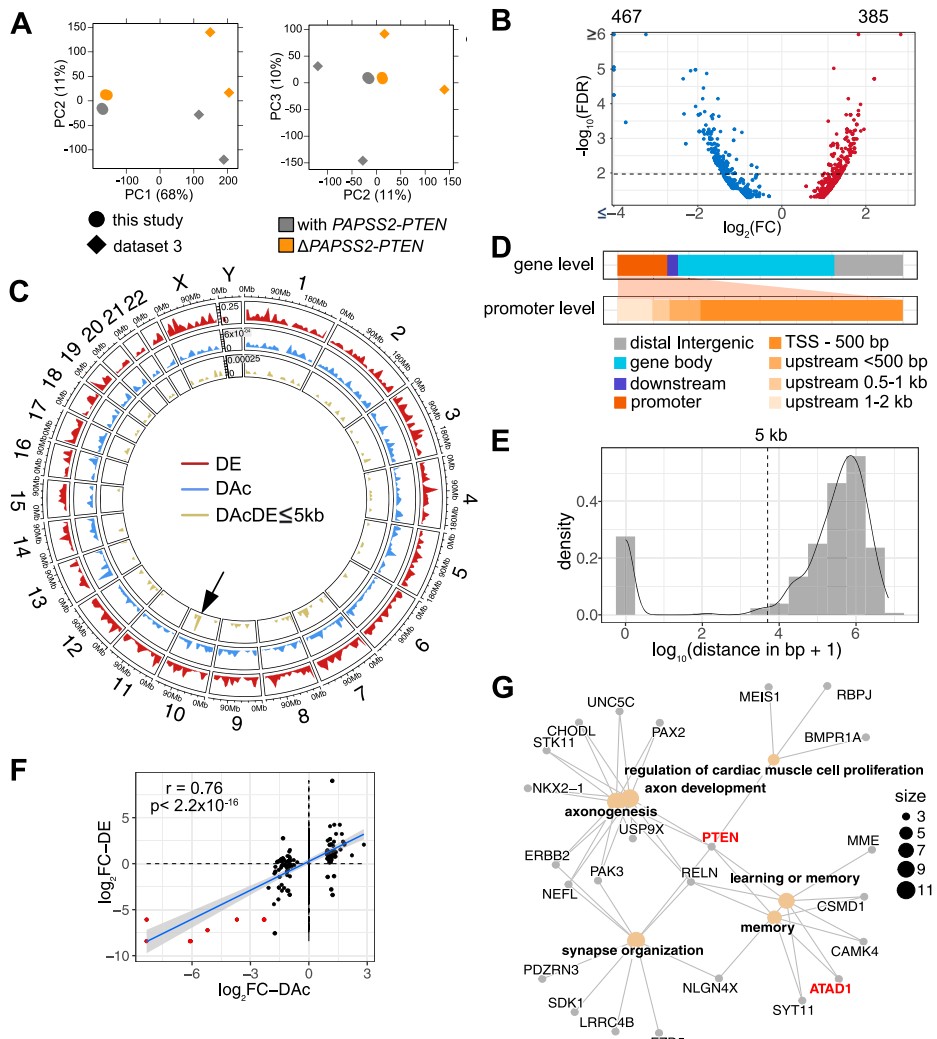

**Figure 4. H3K27ac occupancy was altered in the genomes of PAPSS2-PTEN–deleted HAP1 cells.**
**(A)** Factorial maps of the principal component analysis of global H3K27ac ChIP-seq signals separates samples according to absence (orange) or presence (grey) of the *PAPSS2-PTEN* locus in HAP1 CRISPR-Cas9 deletion clones generated in this (circle) or other (diamond) studies. The proportion of variance explained by each PC is indicated in parenthesis. **(B)** Volcano plot shows differentially acetylated H3K27ac ChIP-seq peaks (DAc) (dashed line corresponds to FDR = 0.01). Number of decreased (blue) and increased (red) acetylated H3K27 peaks are labelled (top). **(C)** Circular representation of the human genome illustrates each chromosome proportionally scaled to its length. Tracks inserted in the circle show the genomic location and frequency of DE genes (red), DAc genomic regions (blue), and DAc peaks with DE genes located nearby (genomic distance DAc to DE, DAcDE ≤ 5 kb, green). Arrow highlights DAcDE pairs on Chr 10. **(D)** Stacked bar plots demonstrate proportional frequencies of DAc peaks for genomic features (color-coded). Subcategories for promoter features are further divided. **(E)** The density plot shows the distance of individual DAc peaks to the nearest DE gene. The dashed line indicates the 5,000 bp cut-off. **(F)** Dot plot correlates fold changes in DAc peaks and nearby DE genes (distance ≤5 kb). The grey area within the plot represents the 95% confidence interval (linear model). Spearman's rank correlation coefficient (r) and *P*-value (p) are indicated. The DAcDE pairs are colored (red) if located within the deleted 10q23.31 locus. **(G)** The gene concept network shows DE genes with a nearby (≤5 kb) DAc peak grouped into significantly enriched GO terms. The size of the node is determined by the number of DE genes contributing to the enriched GO term.

active promoters and enhancers (Kimura, 2013). We used H3K27ac data obtained from our HAP1 Δt1 (Δ*PAPSS2-PTEN*) and control (with *PAPSS2-PTEN*) cell clones (Fig 1A and B). In addition, we retrieved publicly available H3K27ac data from the HAP1 cell clones ΔSM1 (Δ*PAPSS2-PTEN*) and used ΔSMARCC4 (ΔSM4, with *PAPSS2-PTEN*) as control (Figs 2A and S4A, Table S11). After identifying genomic regions enriched for H3K27ac, we performed a PCA that separated the samples first, by data source (PC1 with 68% variance), and second, by the *PAPSS2-PTEN* genotype (PC2 with 11% variance) (Fig 4A).

Our subsequent differential enrichment analysis uncovered 852 differentially acetylated (DAc) H3K27 regions that were distributed across all chromosomes (Figs 4B and C and S4B). About 17% (147/852) of the DAc peaks resided in promoter regions, mostly within 500 bp downstream from the nearest annotated TSSs (Fig 4D). Over half (54%, 463/852) of the DAc peaks were located within gene bodies and nearly a quarter (24%, 204/852) in intergenic regions (Fig 4D). To define the influence of differential H3K27ac on gene expression in Δ*PAPSS2-PTEN* cells, we calculated the distances between each DAc peak and the closest DE gene (Fig 4E). About 85% (730/861) of the DAc peaks were located distant (>5 kb) from any DE

genes. We restricted our subsequent analysis to the 131 DAc peaks (15%, 131/861) that were adjacent to or overlapping with the DE genes (≤5 kb). These DAcDE pairs were spread across many chromosomes and were enriched on Chr 10 that encompassed the *PAPSS2-PTEN* locus (Fig 4D). Differential H3K27 acetylation and gene expression levels of the DAcDE pairs highly correlated (Fig 4F), likely because of DAc of H3K27 in the DE gene promoters and gene body (Fig S4C). DE genes with altered H3K27ac levels were enriched in six GO terms linked to proliferation, development, and memory (Fig 4G). *PTEN* connected all and *ATAD1* 33% (2/6) of the GO terms.

Thus, irrespective of the actual genomic CRISPR-Cas9 modification, we found that the unintended deletion of the *PAPSS2-PTEN* locus in HAP1 cells was related to dramatic changes on the chromatin and transcript level.

### The generation of CRISPR-Cas9 deletion clones aggravated the loss of the *PAPSS2-PTEN* locus

To explain the cause of the Δ*PAPSS2-PTEN*, we systematically tested individual steps commonly performed when generating CRISPR-

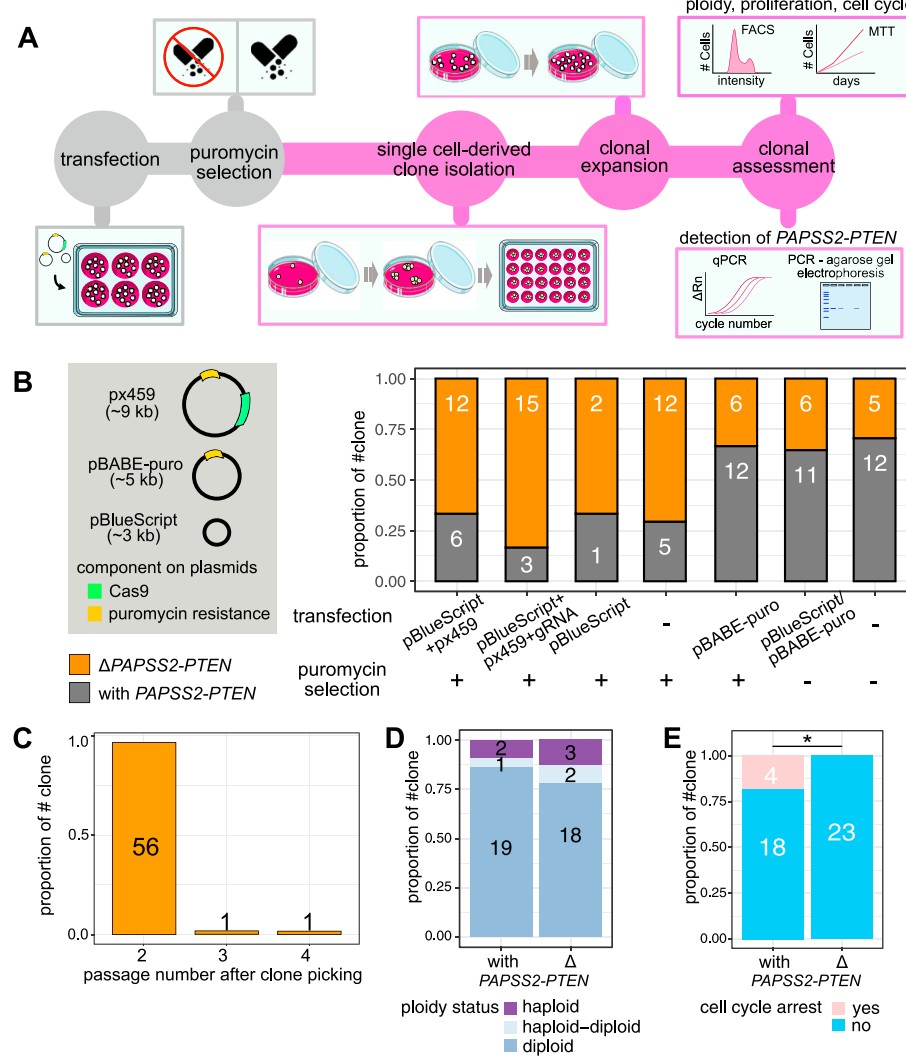

**Figure 5. The generation of CRISPR-Cas9–modified clones exacerbated the loss in the 10q23.31 region resulting in cell cycle changes.**
**(A)** Schematic illustration displays the workflow for determining the frequency of the genomic deletion of the *PAPSS2-PTEN* locus and associated cellular consequences in single-cell–derived HAP1 clones. In some experiments, the transfection and puromycin selection steps (colored in grey) were omitted as part of the testing. **(B, C, D, E)** Bar plots show the frequency of single-cell–derived clones without (orange) and with (grey) the *PAPSS2-PTEN* locus (B) upon transfection with various plasmids (*left*, green: Cas9 gene, yellow: puromycin resistance gene, plasmid size is indicated in kb) with or without puromycin selection, (C) over several cell passages after single cell selection, (D) in different ploidy stages (light blue: haploid, blue: haploid–diploid, purple: diploid) and (E) arrested during cell cycle progression (pink: arrested, blue: not arrested). The number of cell clones for each group is indicated.

Cas9 deletion clones. First, we had already ruled out a gRNA sequence-mediated off-targeting effect because the unintended *PAPSS2-PTEN* deletion was consistently detectable when a variety of different genomic regions were targeted (Fig 2). To study whether Cas9-mediated cleavage in a gRNA sequence-independent manner was required for the deletion of the *PAPSS2-PTEN* locus, we transfected HAP1 cells with a CRISPR-Cas9 plasmid encoding the puromycin resistance gene used for antibiotic-based clonal selection without (px459) or with gRNA sequences (px459 + gRNA_Δt) (Fig 5A and B). We assessed the frequency of the *PAPSS2-PTEN* locus deletion in each single cell-derived clone by PCR or quantitative PCR (qPCR) using primers binding to the *ATAD1* or *PTEN* promoter region (Fig 5A, Table S1). We detected insignificant differences in the frequency of the *PAPSS2-PTEN* deletion in HAP1 cell clones transfected without (67%, 12/18) and with (83%, 15/18) the gRNA sequence-containing CRISPR-Cas9 plasmid (Fisher's exact test, *P* = 0.443) (Fig 5B). This result verified that the genomic deletion of the *PAPSS2-PTEN* locus is independent of gRNA sequences and intracellular gRNA presence, and likely arises during the generation of

CRISPR-Cas9–modified deletion clones and non-targeting control clones.

Second, we co-transfected the large-size CRISPR-Cas9 plasmid with a small-size plasmid (pBlueScript) to enhance transfection efficiency (Søndergaard et al, 2020). We assessed the occurrence of the *PAPSS2-PTEN* deletion in HAP1 cell clones transfected either with or without the small-size plasmid (pBlueScript), which does not encode a puromycin resistance gene. Genomic *PAPSS2-PTEN* locus deletions were detectable in HAP1 cell clones transfected with only the pBlueScript (67%, 2/3) or no plasmid (71%, 12/17) (Fig 5B). This result further corroborated that the *PAPSS2-PTEN* locus deletion in HAP1 cell clones resulted neither through Cas9 off-targeting nor pBlueScript co-transfection.

Lastly, we omitted the delivery of CRISPR-Cas9 components and puromycin selection. Instead, we transfected HAP1 cells with either only the pBlueScript plasmid or no plasmid and propagated the cells in puromycin-free medium. Omitting the antibiotic selection step reduced the occurrences of the *PAPSS2-PTEN* locus deletion in single cell-derived HAP1 clones transfected with the pBlueScript

plasmid (17%, 1/6) or no plasmids (29%, 5/17). Because there was no significant difference in the frequency of the *PAPSS2-PTEN* locus deletion between HAP1 clones transfected with or without small-size plasmid (Fig 5B), we ruled out that the transfection of plasmids affected the *PAPSS2-PTEN* locus deletion.

To investigate when the *PAPSS2-PTEN* locus deletion appeared, we profiled consecutive passages of single cell–derived HAP1 cell clones. We found that the *PAPSS2-PTEN* locus deletion occurred predominantly in the first passage (96%, 45/47) and rarely in the second (2%, 1/47) or third (2%, 1/47) passage (Fig 5C).

HAP1 is considered a near-haploid cell line but frequently transitions to the more stable diploid cell stage (Olbrich et al, 2017; Yaguchi et al, 2018). To investigate whether genome ploidy was linked to the loss of the *PAPSS2-PTEN* locus, we determined the degree of ploidy in 35 single cell–derived HAP1 cell clones with (n = 18) or without (n = 17) the *PAPSS2-PTEN* locus using a mixed population of haploid and diploid HAP1 cells as control. We found that the number of HAP1 cell clones residing in the haploid, haploid-to-diploid-transitioning or diploid stage was almost identical (Figs 5D and S5A–C), verifying that the ploidy status was neither causing nor affecting the deletion of the *PAPSS2-PTEN* locus.

Because PTEN and KLLN have been reported to inhibit cell proliferation, we tested whether the *PAPSS2-PTEN* locus deletion could provide HAP1 cells with a growth advantage. We did not observe a proliferative advantage in Δ*PAPSS2-PTEN* HAP1 cells without exposure to the CRISPR-Cas9 components or puromycin selection (Fig S5A, B, D, and E). However, when exposed to high levels of cellular toxicity, Δ*PAPSS2-PTEN* HAP1 cells were more likely to escape cell cycle arrest, which may give those cells a growth advantage (Figs 5E and S5C and E).

In conclusion, the deletion of the *PAPSS2-PTEN* locus occurred at low frequency in HAP1 cells devoid of plasmid transfections or antibiotic selection. In contrast, the probability of losing the *PAPSS2-PTEN* locus significantly increased during the process of generating CRISPR-Cas9 cell clones and can make HAP1 cells more resilient to cellular stress.

### The *PAPSS2-PTEN* locus deletion in HAP1 cells was evident in human cancers

Because we found that up to 30% of the HAP1 cell clones showed a deletion of the *PAPSS2-PTEN* locus without stressors applied (Fig 5B) and considering that HAP1 cells originated from chronic my-elogenous leukemia, we further investigated the occurrence of this deletion across cancer types using patient data.

First, we inspected gene aberrations of the *PAPSS2*, *ATAD1*, *KLLN*, and *PTEN* loci in 26 cancer types. Overall, we found frequently occurring deep deletions of and mutations in the *PAPSS2* and *PTEN* genes and deep deletions of the *ATAD1* and *KLLN* genes (see the Materials and Methods section). All four genes were deleted in 73% (19/26) of the assessed cancer types (Fig 6A). The frequency of genomic alterations of these loci was relatively low in leukemia, but deep deletions were still highly prevalent. Alterations in cancer genomes are usually large local events, driven by tumor suppressor genes or oncogenes (Bignell et al, 2010; Muller et al, 2012, 2015). When cancer cells lose tumor suppressor genes, the nearby genes

can be collaterally deleted (Bignell et al, 2010; Muller et al, 2012, 2015). We found that the *PTEN* gene locus was lost in 466 patients, many of which had also acquired deep deletions of the *KLLN* (67%), *ATAD1* (59%), and *PAPSS2* (43%) gene locus (Fig 6B). Remarkably, these cumulative gene deletions reflected the gene order in the linear genome, suggesting that *PTEN* is the primary deletion event, and *KLLN*, *ATAD1*, and *PAPSS2* are collaterally deleted (Fig 6B).

Second, we investigated whether the collateral deletion at the *PTEN* locus is accompanied by transcriptional changes. For each cancer type, we searched for matching gene expression and copy number datasets of at least three patients carrying homozygous deletions of these four genes (*PAPSS2*, *ATAD1*, *KLLN*, and *PTEN*) without considering other factors, such as tumor grade, age, or gender of the patients. Of the 23 cancer types, we found four cancer types (prostate adenocarcinoma, lung squamous cell carcinoma, skin cutaneous melanoma, and uterine corpus endometrial carcinoma) with corresponding patient datasets. Based on gene expression, our PCA-separated patient samples according to the occurrence of the *PAPSS2-PTEN* locus deletion in the first three PCs (Fig 6C), which was striking given that other factors would usually distinguish individual patient samples.

Third, we examined to which extent the altered gene expression pattern identified in HAP1 cells was reflected in cancer patients. Genes down-regulated in the Δ*PAPSS2-PTEN* HAP1 cells also showed reduced gene expression in Δ*PAPSS2-PTEN* patient samples (Figs 6D and S6A). For example, 13–45% of the 100 most down-regulated genes in HAP1 were also deregulated in prostate adenocarcinoma, lung squamous cell carcinoma, and skin cutaneous melanoma when comparing patients with and without the *PAPSS2-PTEN* locus deletion (Table S12). Similarly, some of the DE genes which were located within 5 kb of a H3K27 deacetylated site in HAP1 cells also exhibited differences in gene expression in patient samples with and without the *PAPSS2-PTEN* locus deletion (Figs 6E and S6B, Table S12).

Lastly, we observed that the frequency of the *PAPSS2-PTEN* locus deletion significantly increased in prostate cancer patients with radiation therapy (Fig 6F). This finding agreed with our observation that the frequency of the collateral deletion dramatically increased when HAP1 cells were exposed to puromycin in the absence of a resistance gene (Fig 5B).

In summary, the deletion of the *PAPSS2-PTEN* locus resulted in similar gene expression changes across different cancer types and may provide cancer cells with a selective advantage when exposed to external stressors, such as upon cancer treatments.

## Discussion

Frequent genomic alterations, including the deletion at 10q23, have been detected in cancer cell lines (Meléndez et al, 2011; Domcke et al, 2013; Berg et al, 2017). Studying cancer cell lines with known and well-characterized genomic deletions allow to extract essential clinically and pharmacologically relevant information (Meléndez et al, 2011; Domcke et al, 2013; Berg et al, 2017; Wang et al, 2021). However, the 10q23 (Δ*PAPSS2-PTEN*) deletion had remained unnoticed in CRISPR-Cas9–modified HAP1 cell clones. Furthermore, the deletion occurs only in a fraction of HAP1 cell clones resulting in

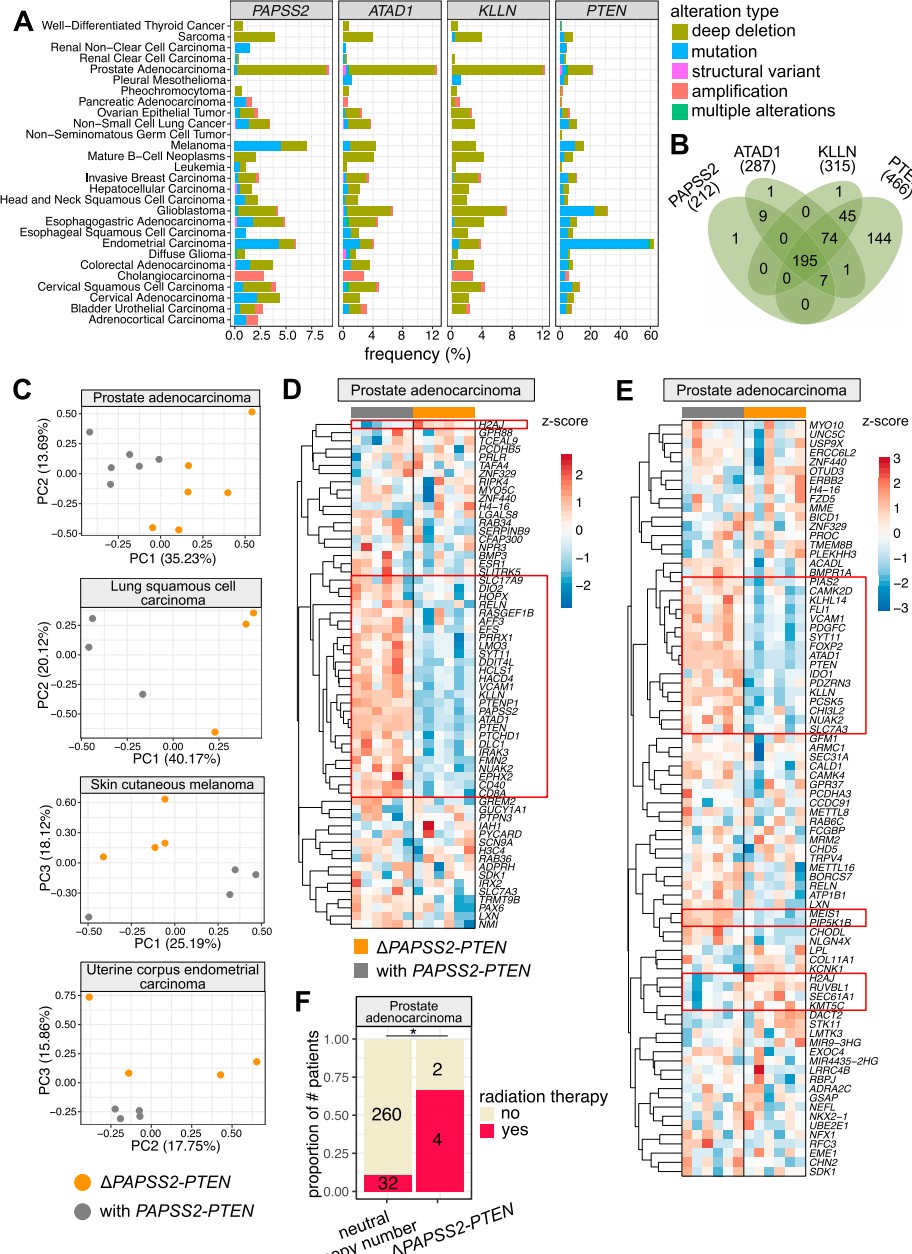

**Figure 6. The deleted *PAPSS2–PTEN* locus occurred frequently in cancer patients.**
**(A)** Bar plots indicate the frequencies of genomic alterations occurring at the *PAPSS2*, *ATAD1*, *KLLN*, and *PTEN* gene loci across 26 human cancer types. The data were retrieved from cBioPortal for Cancer Genomics (Cerami et al, 2012). **(B)** Four-way Venn diagram intersects the numbers of patient samples carrying deep deletions at the *PAPSS2*, *ATAD1*, *KLLN* or *PTEN* gene loci. **(C)** Principal component analysis distinguish patients with (grey) and without (orange) the *PAPSS2–PTEN* gene locus in prostate adenocarcinoma, lung squamous cell carcinoma, skin cutaneous melanoma, and uterine corpus endometrial carcinoma. The proportion of variance explained by each PC is indicated in parenthesis. **(D, E)** Heatmaps show the expression levels of genes in prostate adenocarcinoma patients identified as DE genes in HAP1. The data were log$_{10}$ transformed (red: z-score > 0, blue: z-score < 0). **(E)** The (D) top 100 down-regulated genes ordered by FC and (E) genes with H3K27 DAc sites located within 5 kb are shown. The patient samples on the top of each heatmap are colored by groups (grey: with *PAPSS2–PTEN*, orange: Δ*PAPSS2–PTEN*). Genes with consistent expression changes across the biological replicates are highlighted by red squares. **(F)** Stacked bar plot shows the fraction of prostate adenocarcinoma patients with (neutral copy number) and without the *PAPSS2–PTEN* gene locus upon radiation therapy. Statistics: Fisher's exact test. Significance codes: *0.01 < *P* < 0.05.

a heterogenous pool of HAP1 cell clones with and without the *PAPSS2–PTEN* locus skewing biological observations. Therefore, correct selection of CRISPR-Cas9–modified cell clones with stable genotypes is critical for reliable interpretations of molecular and disease phenotypes.

In addition, many treatments used in cell-based assays are genotoxic and trigger genome instability. For example, antibiotic treatment with puromycin is generally stressful for mammalian cells. As a potent inhibitor of protein synthesis, puromycin will inhibit cell growth in a dose- and exposure time-dependent manner, especially affecting cells without a gene conveying puromycin resistance. Accordingly, we observed that the absence of the puromycin resistance gene increased the frequencies of the

*PAPSS2–PTEN* deletion. However, we also emphasize that up to 30% of all single cell-derived HAP1 cell clones carry the *PAPSS2–PTEN* deletion even without puromycin selection. Thus, despite the px459-mediated puromycin resistance in our CRISPR-Cas9–modified HAP1 cell clones, the increased occurrences of the *PAPSS2–PTEN* deletion in the CRISPR-Cas9–modified cells were likely the consequence of gRNA sequence-independent Cas9 protein-mediated toxicity (Aguirre et al, 2016; Yu et al, 2016; Tycko et al, 2019). Furthermore, Cas9-induced DSB can activate DNA damage checkpoints (van den Berg et al, 2018). Because *PTEN* and *KLLN* encode proteins regulating the G1-S transition (Brandmaier et al, 2017) and cell cycle arrest coupled to apoptosis (Cho & Liang, 2008), respectively, Δ*PAPSS2–PTEN* cells could be faster released

from cell cycle checkpoints. Accordingly, during the puromycin selection process, cells without the resistance gene will be exposed to genotoxic stress (Moran et al, 2009) and thus, the *PAPSS2-PTEN* locus deletion would allow cells to escape cell arrest and consequently survive. The growth advantage achieved through the loss of the *PAPSS2-PTEN* locus under genotoxic stress enabled these cells to outcompete cells with the *PAPSS2-PTEN* locus, allowing them to become dominant in the cell population. We speculate that this proliferation advantage during clonal evolution primarily contributes to the increased frequency of the *PAPSS2-PTEN* deletion in CRISPR-Cas9–modified clones compared with cell clones formed solely through single-cell derivation (Fig 5B). Alternatively, exposure to genotoxic stress might lead to de novo deletion at the *PAPSS2-PTEN* locus in HAP1 cells, resulting in an increased fraction of cells in the population carrying this deletion.

Our findings in HAP1 cell lines underscored the clinical significance of the genomic deletion at 10q23.3 for cancer patient outcome. Previous studies reported the frequent loss of the *PAPSS2-PTEN* locus in prostate cancer and its association with prostate-specific antigen reoccurrence in patients. In addition to impairing the tumor-suppressing roles of PTEN, the ablated metabolic functions of PAPSS2 have been linked to cancer reoccurrences, which emphasize the combined impact of collateral gene deletions in cancer cells (Ibeawuchi et al, 2015; Poluri & Audet-Walsh, 2018). Additive effects of deleted genes were reported in other cancer types and genomic loci as well. For example, glioma cells carrying a large deletion including the *ENO1* gene cannot survive when the paralogue ENO2 was inhibited. Conversely, the depletion of ENO2 only marginally limited cell proliferation when *ENO1* remained intact (Muller et al, 2012). The vulnerability created by collateral deletions in cancer cell provides opportunities for specific and efficient cancer treatment options. Thus, the increase in cancer cell apoptosis upon collateral deletion of genes in the *PAPSS2-PTEN* locus can be exploited in treatment strategies through a targeted co-deletion of *ATAD1* and *PTEN* (Winter et al, 2021 Preprint).

Although the precise mechanism that initiated the *PAPSS2-PTEN* locus deletion remains unclear, our findings indicated mechanistic similarities linked to genome fragility. Genome-wide screening identified hundreds of fragile sites leading to DNA breaks initiated upon cell treatment with DNA replication inhibitors (Debacker & Frank Kooy, 2007; Mrasek et al, 2010). Among them, the *PAPSS2-PTEN* locus resided in the rare fragile site FRA10A (10q23.3). Fragile sites usually give rise to DNA structures that deviate from the classic B-DNA helix, such as R-loops, G-quadruplexes or stem loops, which impede DNA replication and result in replication fork stalling and DNA breaks (Kaushal & Freudenreich, 2019). Previously, it has been shown that *PTEN* exon 1 forms a highly stable secondary structure in vitro (Dillon et al, 2013). In accordance, we identified potential G-quadruplex structures in exonic sequences of *PTEN*, *KLLN*, *ATAD1*, and *CFL1P1* in human cells (Fig S7A) (Lyu et al, 2022). Moreover, our detailed inspection of the sequence content revealed the possible formation of non-B-DNA structures by A-mononucleotide repeat sequences and GT repeats at the DNA break boundaries of the deleted *PAPSS2-PTEN* locus (Fig S7B). We noticed $(A)_{37}$ and $(A)_{30}$ repeat stretches occurring downstream of

*PTEN* gene body that can trigger slipped-strand DNA structures corresponding to single-stranded DNA loops interspersed within double-stranded DNA (Kaushal & Freudenreich, 2019). A-mononucleotide repeat sequences can act as DNA unwinding elements (Bacolla et al, 2016; Tubbs et al, 2018) initiating fork stalling and collapse that results in DNA breaks upon treatment with agents inhibiting DNA synthesis (Tubbs et al, 2018). We found $(GT)_{20}$ repeats located in the first intron of *PAPSS2* that can result in left-handed Z-DNA and induce DSBs in mammalian cells (Wang et al, 2006). Altogether, we reasoned that the *PAPSS2-PTEN* locus is highly vulnerable upon exposure to DNA replication stress.

HAP1 cells were generated as a by-product of transfecting KBM-7 cells with Yamanaka factors to obtain iPSCs (Carette et al, 2011). The two Yamanaka factors c-MYC and KLF4 are proto-oncogenes. Oncogenes are known to generate replication stress (Halazonetis et al, 2008). Moreover, concerns of the genome instability in iPSCs have been raised for years (Blasco et al, 2011). Therefore, we speculate that the exogenous replication stress exerted during the generation of HAP1 cells made them more prone to DSB at the *PAPSS2-PTEN* locus compared with other cell lines. Similarly, DNA replication stress increases considerably during tumor initiation and progression and might explain the high occurrence of *PAPSS2-PTEN* locus deletion in cancer patients.

We found that cancer cells carrying the *PAPSS2-PTEN* locus deletion exhibited significant changes on multiple levels, ranging from global changes in the chromatin environment to cell behavior. This can confound our understandings of cancer cells, especially because HAP1 cells are commonly used in large-scale CRISPR-Cas9–based genetic screens or targeted functional studies of cancer phenotype-associated genes (Sun et al, 2020). Thus, awareness of the dramatic genomic alterations is crucial for CRISPR-Cas9 applications. Furthermore, the molecular characteristics of the *PAPSS2-PTEN* locus deletion that we identified in HAP1 cells can be linked to the commonly observed collateral deletion of these genes in cancer patients. Therefore, our observation underscored the necessity of rigorous HAP1 cell clone validation experiments when applying CRISPR-Cas9–mediated gene-editing experiments, and highlighted the clinical relevance when investigating the impact of collateral gene deletions in cancer patients and the responsiveness to cancer treatments.

# Materials and Methods

### Cell culture

HAP1 cells were purchased from Horizon Discovery with a certified genotype and regularly tested mycoplasma-free. Cells were grown in IMDM, (HyClone) supplemented with 10% FBS, (HyClone), and 1% penicillin–streptomycin (Sigma-Aldrich). Cells were cultured in T75 flasks at 37°C and 5% $CO_2$. HAP1 cells were expanded by splitting 1/10 every 2 d or when reaching 50–60% confluency. Upon splitting, the medium was aspirated, the cells were washed with PBS, (Sigma-Aldrich), and then detached with 2 ml of a trypsin–EDTA solution (Sigma-Aldrich). Trypsin was subsequently inactivated by adding a minimum of threefold surplus of the medium.

## Plasmid construction, transfection, and generation of single-cell–derived clones

gRNAs were designed and the targeting potential was assessed (https://zlab.squarespace.com/guide-design-resources). Each gRNA was individually cloned into the two BbsI restriction sites in pSpCas9(BB)-2A-Puro (px459). For generating non-targeting control clones, px459 without any gRNA sequence was transfected. To enhance transfection efficiency, pBlueScript was co-transfected with the CRISPR-Cas9 plasmid at a 1:1 ratio (Søndergaard et al, 2020). One day before transfection, about 160,000 HAP1 cells were plated in each well of a sixwell plate, and on the next day, the cells were transfected using TurboFectin 8.0 (OriGene) according to the manufacturer's instructions. The cells were incubated together with the plasmid-TurboFectin mixture for 24 h and then selected by adding fresh cell culture medium with 2 µg/ml puromycin for 48 h. Afterwards, cells recovered in the complete medium without puromycin. After recovery, about 100–500 cells were seeded into 10-cm dishes to form single-cell–derived clones. Those clones were hand-picked under the microscope and expanded.

## Genomic DNA extraction

Cells were lysed in 400 µl lysis buffer (0.5% SDS, 0.1 M NaCl, 0.05 M EDTA, 0.01 M Tris–HCl, 200 µg/ml proteinase K). After overnight incubation at 55°C, 200 µl of 5 M NaCl were added, and the sample was vortexed and incubated on ice for 10 min. After centrifugation (15,000$g$, 4°C, 10 min), 400 µl of the supernatant were transferred to a new tube and mixed with 800 µl of 100% ethanol. The samples were incubated on ice for at least 10 min. Genomic DNA was pelleted by centrifugation (18,000$g$, 4°C, 15 min), washed once with 70% ethanol, and resuspended in nuclease-free water.

## PCR

PCR primers were designed using NCBI Primer-BLAST with default parameters. PCR was performed with Taq polymerase (New England Biolabs) according to the manufacturer's instructions. The 25 µl PCR reaction contained 1× standard Taq reaction buffer, 200 µM dNTPs, 0.2 µM primers (Table S1), 1U Taq DNA polymerase, and 100–1,000 ng of genomic DNA. PCR was completed after an initial denaturation step (95°C for 5 min), 35 amplification cycles (95°C for 30 s, 55–60°C for 30 s or for at 68°C for 60 s per 1 kb DNA), a final extension step (68°C for 5 min), and holding the reaction at 4°C in a thermocycler (Applied Biosystems) with a preheated lid (105°C). The products were assessed by 1.2% agarose gel electrophoresis.

## qPCR

Genomic template DNA was incubated at 37°C to homogenously resuspend. About 10–100 ng genomic DNA, 2.5 µM primers targeting the promoter region of *PTEN* or *ATAD1*, and a genomic region on Chr 12 as an internal control (Table S1) and PowerUp SYBR Green Master Mix (Applied Biosystems) were mixed. qPCR was performed using an initial denaturation step (50°C for 2 min,

95°C for 2 min), 40 amplification cycles (95°C for 15 s, 60°C for 1 min), and a step for obtaining the melting curve (95°C for 15 s, 60°C for 1 min and ramp rate 1.6°C/sec, 95°C for 15 s and ramp rate 0.075°C/sec) in a QuantStudio5 Real-Time PCR System (Thermo Fisher Scientific).

## Ploidy and cell cycle assessment

Ploidy level and cell cycle were assessed by flow cytometry (BD LSR II SORP with BD FACSDiva software version 9.0; BD Biosciences) by gating live cells in a FSC-A/SSC-A plot and singlets in a FSC-H/FSC-A plot. DNA content was measured with a 561 nm laser. The results were further analyzed with the FlowJo software version 8.2 and the Dean–Jeff–Fox algorithm for cell cycle analysis and visualization.

About 500,000 cells were collected and washed once in PBS. Cells were fixed by adding 500 µl of ice-cold 70% ethanol drop-wise while vortexing, and subsequently stored at –20°C until further processing. For the cell staining, fixed cells were pelleted and washed twice with PBS and resuspended in 300 µl hypotonic buffer (1 g/liter sodium citrate buffer, 0.1% Triton-X 100) supplemented with 40 µg/ml propidium iodide (Sigma-Aldrich) and 100 µg/ml of RNase A (Thermo Fisher Scientific). A haploid and a diploid cell population were used as gating reference.

## Cell proliferation assay

About 1,000 cells per well were seeded in sextuplets in a 96-well plate, and the cell number was measured daily over 4 d (day 0, 1, 2, and 3). Upon measurement, the culture medium was aspirated, and cells were incubated with a mixture of 60 µl of IMDM and 10 µl of MTT (4 mg/ml Methylthiazolyldiphenyl-tetrazolium bromide, Sigma-Aldrich, in PBS) at 37°C for 75 min. Next, the supernatant was replaced by 100 µl lysis buffer (90% isopropanol, 0.5% SDS, 0.04N HCl) and cells were incubated on a rocker at RT for 30 min. Lysed cells were resuspended and measured on a plate reader (Molecular Devices Spectramax i3x, 595 nm absorbance). Wells without any seeded cells were used as background control. To obtain the optical density (OD), background values were subtracted from the obtained signals per well with seeded cells.

## ChIP-seq

ChIP-seq experiments were performed as previously described (Kutter et al, 2011; Rudolph et al, 2016). Briefly, 15–20 million cells were fixed (1% formaldehyde), lysed, sonicated (Covaris ME220, milliTUBE 1 ml AFA Fiber, with parameter: Setpoint Temperature 9°C, Peak Power 75, Duty Factor 15%, Cycles/Burst 1,000, Duration: ~20 min) and then incubated with H3K4me3 (05-1339; Millipore) and H3K27ac (ab4729; Abcam) antibodies. After ChIP, sequencing libraries were generated (Takara SMARTer ThruPLEX DNA-seq Kit following the manufacturer's protocol), and the library quality and the size distribution were assessed (Agilent Bioanalyzer, High Sensitivity DNA chips) and quantified (KAPA SYBR FAST qPCR kit; Roche). Libraries were sequenced single-end on a NextSeq 500 instrument (Illumina) using the NextSeq 500/550 High Output v2 kit for 75 cycles (Illumina).

## ChIP-seq data analysis

Read quality was assessed by FastQC (Andrews et al, 2015). Reads were aligned to the human reference genome (hg38) using BWA (Li & Durbin, 2009). After alignment, PCR duplicates and reads mapping to the ENCODE exclusion list (https://www.encodeproject.org/files/ENCFF356LFX/) were removed using SAMtools (Li et al, 2009) and NGSUtils (Breese & Liu, 2013). The bam files were indexed and sorted by SAMtools. MACS2 was used for peak calling (Zhang et al, 2008). Subsequently, differential enrichment analysis was performed with DiffiBind (Stark & Brown, 2011). Significantly DAc peaks (FDR ≤ 0.05) were identified. The distances between individual DAc peaks to the nearest DE genes and individual DE genes to the nearest DAc peaks were calculated using BEDTools (Quinlan & Hall, 2010). For visualization, bedgraph files were generated using deepTools (Ramírez et al, 2016). Reads with a mapping quality below 20 were removed with SAMtools, and bedgraph files were visualized with IGV (Robinson et al, 2011).

## RNA extraction and DNase treatment

Cells were harvested at 50–60% confluency. Between 1,000,000 and 5,000,000 cells were pelleted and lysed in 700 μl Qiazol (QIAGEN). Afterwards, 140 μl chloroform were added. The mixture was shaken for 30 s and incubated for 2.5 min at RT, followed by centrifugation (9,000g, 4°C, 5 min) to achieve phase separation. The upper aqueous phase was carefully transferred to a new tube, and one volume isopropanol was added. The tubes were inverted five times to mix thoroughly, followed by incubation (RT, 10 min). The RNA was pelleted by centrifugation (9,000g, 4°C, 10 min) and washed once with ice-cold 70% ethanol. The RNA was resuspended in 30–50 μl nuclease-free water. RNA concentration and purity were determined (2000c; NanoDrop). To remove genomic DNA, 10 μg RNA were mixed with 5 μl 10×TurboDNase buffer (Invitrogen), 1 μl TurboDNase (Invitrogen), 1 μl RNase Inhibitor (RiboLock; Invitrogen), and water wer added to a total volume of 50 μl. The samples were incubated (37°C, 30 min) and purified with the Zymo RNA Clean & Concentrator Kit (Zymo Research). The concentration of purified RNA was determined (RNA HS Assay Kit; Qubit), and the integrity was assessed (RNA 6000 Nano kit; Agilent Bioanalyzer).

## RNA-seq

The RNA was first enriched for molecules with PolyA tails using NEBNext Poly(A) mRNA Magnetic Isolation Module. The RNA library preparation was performed with NEBNext Ultra II Directional RNA Library Prep Kit for Illumina according to the manufacturer's instruction. Library quality was determined (High Sensitivity DNA chips; Agilent Bioanalyzer) and quantified (KAPA SYBR FAST qPCR kit; Roche). Libraries were sequenced paired-end on a NextSeq 500 machine (Illumina) using the NextSeq 500/550 High Output v2 kit for 150 cycles (Illumina).

## RNA-seq data analysis

Sequencing read quality was assessed with FastQC. Adaptor sequences and low-quality reads were trimmed using Trimmomatic (Bolger et al, 2014). Sequencing reads that could be aligned (HiSAT2) (Kim et al, 2019) to annotated ribosomal RNA genes were discarded. Subsequently, the filtered reads were aligned to the human genome hg38 using HiSAT2. Using sorted and indexed bam files, the number of aligned reads was counted for each annotated transcripts (featurecount in subread package) (Liao et al, 2014). For visualization, bedgraph files were generated using deepTools and soft-clipped reads were removed by SAMtools. The raw count tables were used to identify differentially expressed genes (FDR ≤ 0.05) with DESeq2 (Love et al, 2014). The over-representative enrichment of Gene ontology (GO) term and Kyoto Encyclopedia of Genes and Genomes pathway analyses were performed and visualized using ClusterProfiler (Yu et al, 2012).

## RNA-seq and ChIP-seq data selection from public data

In each of the three published datasets used, we found only one CRISPR-Cas9–modified clone carrying *PAPSS2-PTEN* locus deletion (Fig 2A, Tables S2 and S11), which we classified as "Δ*PAPSS2-PTEN*" group.

For the clone selection for the "with *PAPSS2-PTEN*" group, we considered (1) availability of the RNA- and ChIP-seq data in each dataset and (2) the modification of gene expression or protein abundance.

RNA-seq data: (1) In dataset 1, data for only one CRISPR-Cas9-unmodified and one CRISPR-Cas9-modified knockout HAP1 cell clone (*METAP1*, ΔM) were available. Because the CRISPR-Cas9–unmodified HAP1 cell maintained the *PAPSS2-PTEN* locus, we used two replicates. (2) In dataset 2 and 3, data were available for more than one genotypic clone with an intact *PAPSS2-PTEN* locus, including CRISPR-Cas9–unmodified HAP1 clones. However, if we had selected CRISPR-Cas9–unmodified HAP1 clones from all the three datasets, we would have been unable to discern transcriptional changes induced by the deletion of the *PAPSS2-PTEN* locus or the CRISPR-Cas9–modified gene locus and CRISPR-Cas9–associated cellular toxicity. Therefore, instead of choosing unmodified HAP1 clones, CRISPR-Cas9–modified clones with an intact *PAPSS2-PTEN* locus were included (Tables S2 and S11).

ChIP-seq data: Similar to our selection of RNA-seq data, we did not select CRISPR-Cas9–unmodified HAP1 cell clones from the published datasets. There were no matching H3K27ac ChIP-seq and RNA-seq data available for the HAP1 cell clone that we selected. Therefore, we used data of CRISPR-Cas9-modified HAP1 cell clones carrying the dTAG system. In dataset 3, inactivated and activated dTAG systems served as replicates in our ChIP-seq analysis.

## Patient cohort analysis

Publicly available cancer patient metadata from the cancer genome atlas (TCGA) were downloaded via the cBio Cancer Genomics Portal on 25 March, 2022 (Cerami et al, 2012) to retrieve the frequencies of genomic alterations occurring at *PAPSS2*, *ATAD1*, *KLLN*, and *PTEN* and their gene expression levels across cancer types. We intersected patient IDs with deep deletions (−2) in the *PAPSS2*, *ATAD1*, *KLLN*, and *PTEN* genes based on copy number variation (as described on cBioPortal https://docs.cbioportal.org/user-guide/faq/#what-do-amplification-gain-deep-deletion-shallow-deletion-and–

2–1-0-1-and-2-mean-in-the-copy-number-data). We further considered data containing gene expression and copy number information of each patient for the transcriptome analyses. Genes with the average normalized read count smaller than or equal to 10 across all samples were removed. Patient IDs were assigned to the "ΔPAPSS2-PTEN" group having a gene copy number of "-2" for *PAPSS2*, *ATAD1*, *KLLN*, and *PTEN* and a summed up normalized read count smaller than 400 (RNA-seq) (Table S13). In contrast, when *PAPSS2*, *ATAD1*, *KLLN*, and *PTEN* had a gene copy number of "0" and a summed up normalized read count larger than 3,000, we categorized those patient samples into the "with *PAPSS2-PTEN*" group. To match the sample size of the ΔPAPSS2-PTEN group, we randomly sampled patient data from the *PAPSS2-PTEN* group (Table S13).

### Additional data sources

Gene expression data across different cell lines (Fig 1C) were retrieved from the Human Protein Atlas (v. 21.0) (https://www.proteinatlas.org/about/download) under the section RNA HPA cell line gene data (Uhlen et al, 2017). Published Hi-C data on HAP1 cells (Fig 1D) were visualized using 3D Genome browser (Haarhuis et al, 2017; Wang et al, 2018). Published ChIA-PET data for multiple human cell lines (Fig S2C) were visualized through WashU Epigenome Browser (Li et al, 2019). Interaction networks (Fig S3A) were generated with STRING database V11.5 (Szklarczyk et al, 2021). G-quadruplex site predictions and structure mapping using CUT&Tag in HEK293T cells (Fig S7A) were performed (Lyu et al, 2022).

# Data Availability

Scripts used for bioinformatics analyses are available on GitHub: https://github.com/KeyiG/HAP1_10q23_P-Pdel.git. All raw and processed sequencing data generated in this study have been submitted to ArrayExpress (https://www.ebi.ac.uk/arrayexpress/), under accession numbers: H3K4me3 and H3K27ac ChIP-seq data: E-MTAB-11859. RNA-seq data: E-MTAB-11858.

# Supplementary Information

# Acknowledgements

We would like to thank the groups of C Kutter, Marc Friedländer, and Vicent Pelechano, and Ian Mills, Laura Baranello, Marcel Van Vugt, Philip Yuk Kwong Yung, Jing Lyu, and Bruno Urién for the critical discussion and feedback. We thank the Swedish Bioinformatics Advisory Program and Erik Fasterius at the National Bioinformatics Infrastructure Sweden for advice. This work was supported by Chinese Scholarship Council (201700260271 K Geng and C Kutter), EMBO Postdoctoral Fellowships (EMBO-ALTF-1046-2019, EK Brinkman), Knut & Alice Wallenberg Foundation (KAW 2016.0174, C Kutter), Ruth & Richard Julin Foundation (2017-00358, 2018-00328, 2020-00294 C Kutter), SFO SciLifeLab Fellowship (SFO_004, C Kutter), KI KID (2018-00904, C Kutter), Swedish Research Council (2019-05165, C Kutter), Lillian Sagen & Curt Ericsson Research Foundation (2021-00427, C Kutter), Gösta Milton's Research Foundation (2021-00527, C Kutter), and Cancerfonden (22 2246 Pj, C Kutter). Computations were enabled by resources in projects SNIC 2017/7-261, SNIC 2020/16-223, SNIC 2020/15-292, SNIC 2021/22-899, SNIC 2021/23-652, SNIC 2022/22-1063, and SNIC 2022/23-546 provided by the Swedish National Infrastructure for Computing at UPPMAX.

## Author Contributions

K Geng: conceptualization, data curation, software, formal analysis, supervision, validation, investigation, methodology, and writing—original draft, review, and editing.
LG Merino: data curation, formal analysis, validation, investigation, visualization, methodology, and writing—original draft, review, and editing.
RG Veiga: data curation, software, formal analysis, investigation, visualization, methodology, and writing—original draft, review, and editing.
C Sommerauer: data curation, formal analysis, and writing—original draft, review, and editing.
J Epperlein: validation, investigation, methodology, and writing—original draft, review, and editing.
EK Brinkman: data curation, formal analysis, methodology, and writing—original draft, review, and editing.
C Kutter: conceptualization, resources, data curation, formal analysis, supervision, funding acquisition, investigation, visualization, methodology, project administration, and writing—original draft, review, and editing.

## Conflict of Interest Statement

The authors declare that they have no conflict of interest.

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
