## [Reviewer comments · Life Science Alliance]

Life Science Alliance

Intrinsic deletion at 10q23.31, including the PTEN gene locus, is aggravated upon CRISPR Cas9-mediated genome engineering in HAP1 cells mimicking cancer profiles

Keyi Geng, Lara Merino, Raúl Veiga, Christian Sommerauer, Janine Hoffmann, Eva Brinkman, and Claudia Kutter
DOI: <https://doi.org/10.26508/lsa.202302128>

Corresponding author(s): *Claudia Kutter, Karolinska Institutet*

Review Timeline:

Submission Date:	2023-05-02
Editorial Decision:	2023-06-14
Revision Received:	2023-09-28
Editorial Decision:	2023-10-20
Revision Received:	2023-11-06
Accepted:	2023-11-07

Transaction Report:

June 14, 2023

Re: Life Science Alliance manuscript #LSA-2023-02128-T

Claudia Kutter
Karolinska Institute
-
- -
Sweden

Dear Dr. Kutter,

Thank you for submitting your manuscript entitled "CRISPR-Cas9-mediated genome engineering aggravates deletion at 10q23.31 including the PTEN gene locus mimicking cancer profiles" to Life Science Alliance. The manuscript was assessed by expert reviewers, whose comments are appended to this letter. We invite you to submit a revised manuscript addressing the Reviewer comments.

Thank you for this interesting contribution to Life Science Alliance. We are looking forward to receiving your revised manuscript.

Sincerely,

B. MANUSCRIPT ORGANIZATION AND FORMATTING:

Reviewer #1 (Comments to the Authors (Required)):

In their research article, Geng et al. made an unexpected discovery regarding a 287 kb region on Chromosome 10 (10q23.31) in chronic myelogenous leukemia HAP1 cells, commonly used in CRISPR screens. They found a deletion of this region consisting of four protein coding genes: PAPSS2, ATAD1, KLLN, and PTEN. Interestingly, the deletion was not attributed to CRISPR/Cas enzymatic activity but was rather generated or promoted during the CRISPR-Cas engineering process itself. The deletion had significant consequences, as it led to global changes in histone acetylation and gene expression, affecting vital cellular processes like the cell cycle and DNA replication. Notably, similar deletions were detected in cancer patient genomes, resulting in comparable gene expression patterns despite individual differences. As a result, the authors concluded that this unintended deletion of 10q23.31 could significantly impact and confound CRISPR-Cas9 studies.

Comments on the scientific article:

The article highlights an often neglected aspect that clone-specific genome alterations can contribute significantly to the manifestation of clonal phenotypes. This finding has implications for various studies utilizing CRISPR/Cas-dependent high-throughput screening experiments, clonal behavior assays, and copy number variation studies in cancers. While the title strongly claims that "CRISPR-Cas9-mediated genome engineering aggravates deletion at 10q23.31," it is important to consider that this claim addresses only one possible cause for the worsening of the deletion. An alternative explanation could be that the deletion occurred in a subset of HAP1 cells. This is supported by the observation of the same deletion at a lower frequency during puromycin selection and transfection, as described on page 16 (lines 545 onwards). Furthermore, in Figure 1B, the similarity in lack of H3K4me and H3K27Ac between delta T1 and dealt I 17 clones indicates that the deletion was likely a clonal event present in the HAP1 cell population. To support the authors' claim, it would be essential to discuss the possibility of clonal evolution or provide evidence for the presence or absence of the clonal deletion in HAP1 cells.

Minor comments:

1. While the conclusion that the chr10 deletion was not determined by the sgRNA is based on different targeting sequences used, it is important to consider the possibility of off-target activity by each sgRNA. It would be beneficial for the authors to investigate the presence of off-target sites during sgRNA design or provide evidence that no homologous sequences are found at the Chr10 deletion boundaries.
2. The confirmation of the deletion in other HAP1 CRISPR/Cas- clones, occurring in 44-61% of the clones while not being detected in control clones, does not rule out the possibility of a clonal effect. It would be helpful for the authors to investigate whether the deletion occurs in a small population of cells within the bulk HAP1 cell population.
3. It is unclear whether the unmodified clones and dt-dt1 clones are related. Providing information on the junction sequences at the 10q23.31 deletion and their clonality would be beneficial for better understanding the findings.
4. The authors observed that HAP1 cells show increased resistance to external stress and overall replication stress. It would be informative for the authors to demonstrate whether this phenotype is primarily dependent on PTEN or other genes that are deleted. e.g. overexposes PTEN in some of the HAP1 clones see if it reverts the epigenetic, or gene expression profiles.

Reviewer #2 (Comments to the Authors (Required)):

SUMMARY:

This study reports the inconsistent occurrence of a large and unexpected genomic deletion at 10q23.31 in HAP1 cells, a widely used human haploid cell model. The 287 kb deleted region on chromosome 10 encompasses several protein coding genes that have regulatory functions including PAPSS2, ATAD1, KLLN and PTEN, as well as the noncoding pseudo-gene CFL1P1 (i.e. cofilin-like). The authors report that deletion of this region only occurred in CRISPR/Cas9 edited clones, specifically when they generated variants of two proximal transfer RNA genes on chromosome 17. Deletion of the 10q23.31 region was also found to be associated with global changes in histone acetylation and gene expression, which can alter the internal wiring of a cell and confound biological studies. A key claim in the manuscript is that generation of CRISPR-Cas9 edited cells and/or exposure to

cellular stressors greatly increases the occurrence of the 10q23.31 deletion. They also provide evidence that the 10q23.31 deletion is a collateral deletion that occurs frequently in patient tumors.

CRITIQUE:

The observation of 10q23.31 collateral deletion in some clonal HAP1 cell lines is noteworthy, given that HAP1 cells are a commonly used model human cell line. However, mutation and deep deletion at the PTEN locus among patient tumors is quite common and well established. The tendency for co-occurrence of alterations for all combinations of PAPPSS2, ATAD1, KLLN and PTEN is highly significant across several different cancer types (cbioportal.org). It is also well known that loss of PTEN can cause abnormal transcript signatures linked to cell cycle and DNA replication phenotypes. Better integration with previously published work reporting the effects of PTEN loss on transcriptome outcomes would elevate the manuscript.

The importance of this study is in the proposed mechanism and frequency by which the PAPPSS2-PTEN deletion occurs, which may have broader impacts for engineering cell lines. Obviously collateral PTEN loss could have dire consequences for cell-based therapies. For example, autologous CAR-T based therapies that may incorporate a gene editing step with CRISPR/Cas9 to introduce a beneficial mutation or even the CAR construct itself at a defined locus. To explain the cause of the PAPPSS2-PTEN deletion, the authors systematically tested individual steps commonly performed when generating CRISPR/Cas9 deletion clones and ruled out gRNA effects, Cas9 off-targeting, pBluescript co-transfection, transfection of plasmids in general, and ploidy status. The state that "the deletion of the PAPPSS2-PTEN locus occurred at low frequency in HAP1 cells devoid of plasmid transfections or antibiotic selection" and "the probability of losing the PAPPSS2-PTEN locus significantly increased during the process of generating CRISPR-Cas9 cell clones and can make cells more resilient to cellular stress". Unless I was misunderstanding Figure 5B, it looks like the authors are showing that puromycin selection causes a much higher proportion of PAPPSS2-PTEN cells compared with no puromycin treatment. At a minimum, the authors should consider doing the following:

- [1] See if the PAPPSS2-PTEN deletion occurs with other selection markers (e.g. blasticidin, neomycin, hygromycin, zeocin), and compare the frequency of occurrence.
- [2] If the authors are correct and HAP1 cells are prone to a high frequency of PAPPSS2-PTEN cells following genome engineering, the authors should use a different technology to demonstrate that this indeed the case (e.g. Zn finger nucleases or TALENs).
- [3] The authors should attempt to report on the frequency of PAPPSS2-PTEN deletions in other commonly used cell lines (e.g. RPE1-hTERT, HEK293T, etc..).

In addition to these experiments, do the authors know if TP53 status influences the frequency of PAPPSS2-PTEN deletion clones in any given population of cells? What would happen in HAP1 cells if you restored them to carry the wild-type allele of TP53?

The authors also state that PAPPSS2, ATAD1, KLLN and PTEN "exert essential molecular functions" (line 324) and "Given that this region is important for 3D genome organization and encodes four protein coding genes with essential molecular functions, its unintended deletion may dominate over the expected gRNA-mediated CRISPR/Cas9 genomic deletion and could lead to biological misinterpretations" (lines 335-338). In fact, none of these genes are classified as common or selectively essential genes in the Cancer Dependency Map data so the authors should clarify which specific essential functions they are referring to. Although this reviewer agrees that the deletion could confound biological interpretation, the authors could also provide some concrete examples of how this could occur.

Lastly, the authors show that clones harboring the PAPPSS2-PTEN deletion have altered transcriptional profiles, as well as H3K4me3 and H3K27ac profiles. When the authors looked at published RNA-seq datasets where various genes were mutated with CRISPR-Cas9, they did not find any direct interactions with the PAPPSS2-PTEN deleted genes, but after performing a principal component analysis, the principal components PC3-PC5 separated samples based on PAPPSS2-PTEN presence or absence. It would be interesting to see whether re-expression of 1 or a combination of the deleted genes restores the transcriptional and epigenetic profiles in HAP1 cells. Moreover, starting at line 590 they describe that PAPPSS2-PTEN genes are deleted in many cancer cell types, but that PTEN is deleted in the highest number. The deletion of the other genes in this locus (i.e. ATAD1, KLLN and PAPPSS2) appears to occur in an accumulative manner following the gene order in the linear genome. Could it be that PTEN drives the majority of the transcriptional and epigenetic profile changes in Cas9-modified HAP1 cells. What would happen if you started the experiment with a PTEN mutant HAP1 cell line? Would this be less prone to PAPPSS2-PTEN occurrences? From the clinical data they examine, they find that "deep deletion" occurs commonly in prostate adenocarcinoma cells. It would be informative to see whether CRISPR/Cas9 modification of prostate cancer cell models are particularly sensitive to increased PAPPSS2-PTEN deletion rate relative to other cell types. And whether this is true of other cancer types where PTEN deletions are commonly found.

Reviewer #3 (Comments to the Authors (Required)):

The paper by Geng et al., describes an interesting finding that CRISPR-Cas9-modified HAP1 cells obtained an unexpected

deletion of 10q23.31 region that contains four recurrently deleted genes in human cancers, including the tumor suppressor gene PTEN. The authors initially observed 10q23.31 loss from chromatin profiling data in CRISPR-Cas9-modified HAP1 cells, and further investigated 10q23.31 loss in a panel of diverse human cancer patient cells using public databases. The authors found that 10q23.31 loss caused significant changes in both H3K27ac occupancy and transcription level of a number of cell cycle and DNA replication related genes. The altered transcription signature was also validated using public human cancer patient databases. In an effort to elucidate the mechanism involved in generating 10q23.31 loss in CRISPR-Cas9-modified HAP1 cells, the authors concluded that the process of CRISPR-Cas9-mediated gene editing triggered 10q23.31 loss, and genotoxic stress potentially enhanced the frequency of 10q23.31 loss. In line with this, radiotherapy induced more 10q23.31 loss in clinical. Although the mechanism of 10q23.31 loss in cancer patient cells is still elusive, this paper carefully explored and discussed multiple potential factors during CRISPR-Cas9-mediated gene targeting that could affect the frequency of 10q23.31 deletion.

Overall, this manuscript demonstrated an important recurrent phenomenon caused by CRISPR-Cas9-mediated gene editing in HAP1 cells, which may represent a general caveat of using CRISPR-Cas9-mediated gene editing in other cell lines as well and reflect the similar genetic event that occurs in diverse cancer patient cells. The authors provided some mechanistic and clinical insights on the recurrent 10q23.31 loss in human cancers. There is already a lot of supportive data for this discovery. However, there are still some concerns when addressed would further strengthen this manuscript.

- 1, Whether the dual gRNA system targeting two proximal transfer RNA (tRNA) genes caused off-target effect to generate $\Delta t1$ and $\Delta i17$ clones? Did the author use two control sgRNA pairs for the CRISPR-Cas9-unmodified control (ctrl) clone?
- 2, The authors made conclusion that "Since the guide RNA sequences (gRNAs) used for generating the Δt and Δi cell clones were different, we concluded that the 10q23.31 deletion occurred in a gRNA-independent manner". The fact that "the guide RNA sequences (gRNAs) used for generating the Δt and Δi cell clones were different" is supportive for conclusion that 10q23.31 deletion occurred in a gRNA sequence-independent manner. If so, the author should be clear about what control gRNAs they used, and the clone with control gRNA pairs should also have the same deletion; in contrast, the parental cells would have no 10q23.31 deletion. The authors should also consider whether gRNA-guided genetic cutting by CRISPR-Cas9 is required for triggering 10q23.31 loss, or transfection of CRISPR-Cas9 is sufficient to cause 10q23.31 deletion?
- 3, The authors found that "Hi-C and ChIA-PET data obtained from HAP1 and other cell types revealed multiple long-range interactions between the 10q23.31 and other genomic regions indicating additional roles in three-dimensional (3D) gene regulation". One wonders whether those gRNAs that triggered 10q23.31 deletion targeted the regions involved in 10q23.31's long-range interactions, thereby, facilitated 10q23.31 deletion during the process of CRISPR-Cas9-induced gene editing/repair?
- 4, The authors showed the chromatin profiling of H3K4me3 and H3K27ac to visualize 10q23.31 deletion. How did Input or IgG control chromatin profiling look like at the same regions?
- 5, "We next examined the top 20 genes contributing to PC1 and PC2. ATAD1, PTEN and PAPSS2 were among the top genes that contributed to the strongest separation of the two groups (Fig. 2D). None of the top genes included the originally indented gene knockouts (Fig. 2D) and were located on different chromosomes (Supplementary Table S4-5)." This part is confusing. ATAD1, PTEN and PAPSS2 were among the top 20 genes and were located on the same chromosome, why the authors claimed that "None of the top genes were located on different chromosomes". In addition, before batch correction, did authors see predominant expression changes induced by intended gene editing? What kind of controls (parental or control gRNAs) the other three datasets used?
- 6, The authors used published RNA-seq datasets of CRISPR-Cas9 edited HAP1 cells from other separate studies. Are these datasets generated from single population clones as well? While the authors used dual gRNA system, one wonders whether the other studies used single gRNA system? If so, what is the authors' thoughts on the effect of different CRISPR-Cas9 methods on the frequency of 10q23.31 deletion. How frequent and how long it takes to generate 10q23.31 deletion upon CRISPR-Cas9 gene editing? And how much this can reflect 10q23.31 deletion in naturally occurred cancer patient cells?
- 7, Whether these differentially expressed genes and H3K27ac changes caused by loss of PAPSS2-PTEN locus is not convincing. Would knockout of PAPSS2-PTEN in control cells or restoration of PAPSS2-PTEN in Δ PAPSS2-PTEN cells mimic or rescue the transcription and H3K27ac changes?
- 8, "This result verified that the genomic deletion of the PAPSS2-PTEN locus was gRNA-independent and likely caused during the generation of CRISPR-Cas9 deletion clones." This is confusing. While the observation that multiple different gRNA with Cas9 triggered PAPSS2-PTEN loss supported that deletion of the PAPSS2-PTEN locus was gRNA sequence-independent, whether or not gRNA-mediated Cas9 cutting was required for PAPSS2-PTEN loss is unclear. The author compared Cas9 without or with gRNA and did not see significant difference in the frequency of PAPSS2-PTEN loss, indicating that gRNA-mediated cutting was not required. This is inconsistent with what the author claimed in the referred statement that "caused by generation of deletion".
- 9, The authors should mention in line 541-544 that puromycin selection was applied to the culture although the cells did not receive puromycin resistance gene. This is different from the next paragraph that puromycin selection was omitted. The number

of single cell clones used for analysis varied a lot from 50 or so to 3. One worried that statistics from those small amounts of samples may cause significant bias to draw conclusion. In addition, one wonders the efficiency of these transfection assays, especially without selection marker, how the authors distinguish transfected versus non-transfected cells and how this would affect the conclusion?

10, Transfection without antibiotic selection had reduced occurrences of PAPSS2-PTEN loss may be due to non-transfected parental cells. Transfection with Cas9-puromycin or puromycin alone with antibiotic selection could determine whether Cas9 itself affects the frequency of PAPSS2-PTEN loss. Why transfection with or without plasmids still can induce PAPSS2-PTEN loss, albeit much lower frequency?

11, Did the authors also see increased frequency of PAPSS2-PTEN loss in drug treated patients or cells? Is PAPSS2-PTEN loss more related to genotoxic stress/exposure compared with non-genotoxic agents?

12, Did the authors observe correlation between oncogene overexpression (eg. Myc) and PAPSS2-PTEN loss in cancer patient samples?

13, some minor comments:

The authors may have mistakes in lines 370-375 regarding dataset 3 of Δ SM1 or Δ SM2. The term gRNA-independent is unclear. It could mean gRNA sequence-independent, but gRNA is required for Cas9-mediated gene editing to induce PAPSS2-PTEN loss, or the authors meant Cas9-mediated cutting without gRNA. The authors may clarify this. Some typos, such as in line 668, 700, etc.

We thank the reviewers for their assessments of our work and their constructive comments. This has helped us design new analysis and experiments to improve the clarity and the impact of our findings to the field, while adhering to the journal's requirements. The original comments are provided, and our point-by-point responses are in red. For the reviewers' convenience, we have inserted the new data in form of figures/table/note for the reviewer (Fig. R/Table. R/Note. R). We trust that all comments have been addressed satisfactorily either through additional analysis/experiments or through clarifying specific points.

Reviewer #1 (Comments to the Authors (Required)):

In their research article, Geng et al. made an unexpected discovery regarding a 287 kb region on Chromosome 10 (10q23.31) in chronic myelogenous leukemia HAP1 cells, commonly used in /Cas9PR screens. They found a deletion of this region consisting of four protein coding genes: PAPSS2, ATAD1, KLLN, and PTEN. Interestingly, the deletion was not attributed to CRISPR/Cas enzymatic activity but was rather generated or promoted during the CRISPR-Cas engineering process itself. The deletion had significant consequences, as it led to global changes in histone acetylation and gene expression, affecting vital cellular processes like the cell cycle and DNA replication. Notably, similar deletions were detected in cancer patient genomes, resulting in comparable gene expression patterns despite individual differences. As a result, the authors concluded that this unintended deletion of 10q23.31 could significantly impact and confound CRISPR-Cas9 studies.

Comments on the scientific article:

The article highlights an often neglected aspect that clone-specific genome alterations can contribute significantly to the manifestation of clonal phenotypes. This finding has implications for various studies utilizing CRISPR/Cas-dependent high-throughput screening experiments, clonal behavior assays, and copy number variation studies in cancers. While the title strongly claims that "CRISPR-Cas9-mediated genome engineering aggravates deletion at 10q23.31," it is important to consider that this claim addresses only one possible cause for the worsening of the deletion. An alternative explanation could be that the deletion occurred in a subset of HAP1 cells. This is supported by the observation of the same deletion at a lower frequency during puromycin selection and transfection, as described on page 16 (lines 545 onwards). Furthermore, in Figure 1B, the similarity in lack of H3K4me and H3K27Ac between delta T1 and dealt I 17 clones indicates that the deletion was likely a clonal event present in the HAP1 cell population. To support the authors' claim, it would be essential to discuss the possibility of clonal evolution or provide evidence for the presence or absence of the clonal deletion in HAP1 cells.

We thank for the reviewer's input on the way we address our conclusion.

1. We have revised our title to "Intrinsic deletion at 10q23.31, including the *PTEN* gene locus, is aggravated upon CRISPR-Cas9-mediated genome engineering in HAP1 cells mimicking cancer profiles" to fully acknowledge that the deletion occurred in a subset of parental HAP1 cells.
2. We have also included discussing the clonal evolution (Discussion, second paragraph): "The growth advantage achieved through the loss of the *PAPSS2-PTEN* locus under genotoxic stress enabled these cells to outcompete cells with the *PAPSS2-PTEN* locus, allowing them to become dominant in the cell population. We speculate that this proliferation advantage during clonal evolution primarily contributes to the increased frequency of the *PAPSS2-PTEN* deletion in CRISPR-Cas9-modified clones compared to cell clones formed solely through single-cell derivation (Fig. 5B). Alternatively, exposure to genotoxic stress might lead to *de novo* deletion at the *PAPSS2-PTEN* locus in HAP1 cells, resulting in an increased fraction of cells in the population carrying this deletion."

Minor comments:

1. While the conclusion that the chr10 deletion was not determined by the sgRNA is based on different targeting sequences used, it is important to consider the possibility of off-target activity by each sgRNA. It would be beneficial for the authors to investigate the presence of off-target sites during sgRNA design or provide evidence that no homologous sequences are found at the Chr10 deletion boundaries.

We initially disregarded the possibility of gRNA-dependent Cas9 off-target editing due to the use of different targeting sequences. Importantly, we subsequently validated this speculation by identifying HAP1 single cell-derived clones carrying this large deletion even in the absence of cellular transfection with CRISPR-Cas9 components (**Fig. 5B**). Furthermore, we also detected the deletion in HAP1 clones obtained upon transfection with diverse gRNA sequences (**Fig. 2A**). Consequently, we concluded that the unexpected genomic deletion was not a direct outcome of Cas9 off-target cleavage. Therefore, we reasoned that this large genomic deletion can be found in any sgRNA design system employed in HAP1 cells.

Nonetheless, we appreciate the reviewer's concern. To provide additional support for our assertion that the deletion is not a result of gRNA-dependent off-target activity, we performed pairwise sequence alignment between each sgRNA and the sequences of the break sites (**Note R1**). For more detailed information of how we determined the sequences of the break sites, please refer to our response to Reviewer 1, Point 3.

gRNA-Δt-1	1	-----CTTAGGA-----AAAATCGACACCC--	20	gRNA-Δt-2	1	-----CTAATGA	7
left_boundary	1	AAGCAGGTTTCAAATCCAGGCTGCTT--GACTCCAATAATCGATACTCAT	48	left_boundary	1	AAGCAGGTTTCAAATCCAGGCTGCTT--GACTCCAATAATCGATACTCATG-	49
gRNA-Δt-1	21	-----	20	gRNA-Δt-2	8	CAACGTAGTGGCG-	20
left_boundary	49	GCTACTCCGATGCATGCTACAGAGAGTTTTCTGGGCCAGTAGTGGGAAG	98	left_boundary	50	CTACTCCGATGCATGCTACAGAGAGTTTTCTGGGCCAGTAGTGGGAAGA	99
gRNA-Δt-1	21	-----	20	gRNA-Δt-2	21	-----	20
left_boundary	99	AAGTAGAGATTTTGCAAAATGTCTTAGGACCCAGTATAGAAGTATG	148	left_boundary	100	AGGTAGAGATTTTGCAAAATGTCTTAGGACCCAGTATAGAAGTATGC	149
gRNA-Δt-1	21	-----	20	gRNA-Δt-2	21	-----	20
left_boundary	149	CCACTAACTTTGGGGTTTCTGGACCACTTCCTGATAAGGGGCAGAGTTAG	198	left_boundary	150	CACTAACTTTGGGGTTTCTGGACCACTTCCTGATAAGGGGCAGAGTTAGA	199
gRNA-Δt-1	21	-----	20	gRNA-Δt-2	21	-----	20
left_boundary	199	ATGGGATGCCGAGGATGCTGTGTGTTCTCTATGTACACACAGCAAGACA	248	left_boundary	200	TGGGATGCCGAGGATGCTGTGTGTTCTCTATGTACACACAGCAAGACAG	249
gRNA-Δt-1	21	-----	20	gRNA-Δt-2	21	-----	20
left_boundary	249	GACATTTTTCCTTTAACAATGTGTTTGTCTTCTATAGAGAGAATCCTTT	298	left_boundary	250	ACATTTTTCCTTTAACAATGTGTTTGTCTTCTATAGAGAGAATCCTTTT	299
gRNA-Δt-1	21	---	20	gRNA-Δt-2	21	--	20
left_boundary	299	TTC	301	left_boundary	300	TC	301
gRNA-Δi-1	1	-----	0	gRNA-Δi-2	1	-----CTCGTTTGTGGTCTCCGG-----	20
left_boundary	1	AAGCAGGTTTCAAATCCAGGCTGCTT--GACTCCAATAATCGATACTCATGC	50	left_boundary	1	AAGCAGGTTTCAAATCCAGGCTGCTT--GACTCCAATAATCGATACTCATGC	50
gRNA-Δi-1	1	-----	0	gRNA-Δi-2	21	-----	20
left_boundary	51	TACTCCGATGCATGCTACAGAGAGTTTTCTGGGCCAGTAGTGGGAAGAA	100	left_boundary	51	TACTCCGATGCATGCTACAGAGAGTTTTCTGGGCCAGTAGTGGGAAGAA	100
gRNA-Δi-1	1	-----	0	gRNA-Δi-2	21	-----	20
left_boundary	101	GGTAGAGATTTTGCAAAATGTCTTAGGACCCAGTATAGAAGTATGCC	150	left_boundary	101	GGTAGAGATTTTGCAAAATGTCTTAGGACCCAGTATAGAAGTATGCC	150
gRNA-Δi-1	1	-----ACTAAGCACTAC-----GTCCGGT-----	20	gRNA-Δi-2	21	-----	20
left_boundary	151	ACTAACTTTGGGGTTTCTGGACCACTTCCTGATAAGGGGCAGAGTTAGAT	200	left_boundary	151	ACTAACTTTGGGGTTTCTGGACCACTTCCTGATAAGGGGCAGAGTTAGAT	200
gRNA-Δi-1	21	-----	20	gRNA-Δi-2	21	-----	20
left_boundary	201	GGGATGCCGAGGATGCTGTGTGTTCTCTATGTACACACAGCAAGACAGA	250	left_boundary	201	GGGATGCCGAGGATGCTGTGTGTTCTCTATGTACACACAGCAAGACAGA	250
gRNA-Δi-1	21	-----	20	gRNA-Δi-2	21	-----	20
left_boundary	251	CATTTTTCCTTTAACAATGTGTTTGTCTTCTATAGAGAGAATCCTTTT	300	left_boundary	251	CATTTTTCCTTTAACAATGTGTTTGTCTTCTATAGAGAGAATCCTTTT	300
gRNA-Δi-1	21	-	20	gRNA-Δi-2	21	-	20
left_boundary	301	C	301	left_boundary	301	C	301

```

gRNA-Δt-1      1 -----CTTAGGAAAAATC---GAC      16
right_boundar  1 TATTCAGTGTAAACAAGGAATAAGCTATCAACTCAGGAAAAGACATGGAA      50
gRNA-Δt-1     17 ACCC-----CTAATGACAACGT---AGTGGC-----      20
right_boundar  51 GAACATTAGATGCATATGACTGAGTGAAGAAGCCAATCCAAAAAGACTA      100
gRNA-Δt-1     21 -----      20
right_boundar  101 CATACTGTATTACTCCAACATATATGACATTTTGAAAAAGGCCAAAACATATG      150
gRNA-Δt-1     21 -----      20
right_boundar  151 GAGACAGCTTAAAGATCAGTGGTTGCCAGGGATTAGGGGTGAGGAAAGGA      200
gRNA-Δt-1     21 -----      20
right_boundar  201 TGAATCGGTAGGGCACAGAGGATTTTCAGGGCAGTAAAATTATTTCAGTAT      250
gRNA-Δt-1     21 -----      20
right_boundar  251 AACACTATAATGGTAGATACATCATTACACATTTGTCCAACCCACAGCA      300
gRNA-Δt-1     21 -      20
right_boundar  301 T      301

gRNA-Δi-1     1 -----      0
right_boundar  1 TATTCAGTGTAAACAAGGAATAAGCTATCAACTCAGGAAAAGACATGGAA      50
gRNA-Δi-1     1 -----ACTAAAGCACTA      12
right_boundar  51 GAACATTAGATGCATATGACTGAGTGAAGAAGCCAATCCAAAAAGACTA      100
gRNA-Δi-1     13 CGTGGGT-----      20
right_boundar  101 CATACTGTATTACTCCAACATATATGACATTTTGAAAAAGGCCAAAACATATG      150
gRNA-Δi-1     21 -----      20
right_boundar  151 GAGACAGCTTAAAGATCAGTGGTTGCCAGGGATTAGGGGTGAGGAAAGGA      200
gRNA-Δi-1     21 -----      20
right_boundar  201 TGAATCGGTAGGGCACAGAGGATTTTCAGGGCAGTAAAATTATTTCAGTAT      250
gRNA-Δi-1     21 -----      20
right_boundar  251 AACACTATAATGGTAGATACATCATTACACATTTGTCCAACCCACAGCA      300
gRNA-Δi-1     21 -      20
right_boundar  301 T      301

gRNA-Δi-2     1 -----      0
right_boundar  1 TATTCAGTGTAAACAAGGAATAAGCTATCAACTCAGGAAAAGACATGGAA      50
gRNA-Δi-2     1 -----CT-      2
right_boundar  51 GAACATTAGATGCATATGACTGAGTGAAGAAGCCAATCCAAAAAGACTA      100
gRNA-Δi-2     3 CGTTTTGTTGGTCTCCGG-----      20
right_boundar  101 CATACTGTATTACTCCAACATATATGACATTTTGAAAAAGGCCAAAACATATG      150
gRNA-Δi-2     21 -----      20
right_boundar  151 GAGACAGCTTAAAGATCAGTGGTTGCCAGGGATTAGGGGTGAGGAAAGGA      200
gRNA-Δi-2     21 -----      20
right_boundar  201 TGAATCGGTAGGGCACAGAGGATTTTCAGGGCAGTAAAATTATTTCAGTAT      250
gRNA-Δi-2     21 -----      20
right_boundar  251 AACACTATAATGGTAGATACATCATTACACATTTGTCCAACCCACAGCA      300
gRNA-Δi-2     21 -      20
right_boundar  301 T      301

```

Note R1. No homologous sequences are found between gRNA and the deletion boundaries that could induce Cas9 off-target activity.

We used gRNA-Δt-1 and gRNA-Δt-2 to create Δt clones, and gRNA-Δi-1 and gRNA-Δi-2 for generating Δi clones. The term “left_boundary” refers to the sequence located at 5’ end of the deletion boundaries, whereas “right_boundary” denotes the sequence located at 3’ end of the deletion boundaries. For the pairwise alignment we utilized EMBOSS Needle with default parameters (https://www.ebi.ac.uk/Tools/psa/emboss_needle/).

Based on the result of the pairwise alignment (**Note R1**), we can confidently conclude that the genomic deletion was not directly caused by the homologous sequences between the breakage boundaries and the sgRNA.

Furthermore, we performed additional off-target screening of our four gRNAs using the tools Cas-OFFinder (Bae et al. 2014) (<http://www.rgenome.net/cas-offinder/>). The analysis also did not reveal any potential off-target sites located on Chr 10 (**Table R1**). Importantly, most of the predicted off-target sites shown in Table R1 were only found using very lenient parameters allowing for two mismatches and one bulge at the same time. Therefore, we consider this prediction inclusive, and it is highly unlikely that any of our four gRNAs used in our study led to gRNA-sequence dependent off-target sites on Chr 10.

gRNA	Chromosome	Position	Direction
gRNA-Δi-1	chr11	27651274	+
gRNA-Δi-1	chr16	26098963	+
gRNA-Δi-1	chr17	39153817	-
gRNA-Δi-1	chr5	77728748	+
gRNA-Δi-1	chr6	14817908	-

gRNA- Δ i-2	chr1	114138921	+
gRNA- Δ i-2	chr17	39154450	-
gRNA- Δ i-2	chr19	4008855	+
gRNA- Δ i-2	chr22	45337193	+
gRNA- Δ i-2	chr4	153018942	+
gRNA- Δ i-2	chr9	35103112	-
gRNA- Δ t-1	chr1	94806985	+
gRNA- Δ t-1	chr13	99010584	+
gRNA- Δ t-1	chr17	39153680	+
gRNA- Δ t-1	chr17	39153682	+
gRNA- Δ t-1	chr2	166803037	-
gRNA- Δ t-1	chr20	36962872	-
gRNA- Δ t-1	chr4	120680983	+
gRNA- Δ t-1	chr5	15758101	-
gRNA- Δ t-1	chr6	5505814	-
gRNA- Δ t-1	chr9	23639468	+
gRNA- Δ t-1	chrX	112644702	+
gRNA- Δ t-1	chrX	21873073	+
gRNA- Δ t-1	chr13	99010584	+
gRNA- Δ t-1	chr17	39153680	+
gRNA- Δ t-2	chr12	25242652	-
gRNA- Δ t-2	chr12	115512396	-
gRNA- Δ t-2	chr17	39154563	-
gRNA- Δ t-2	chr2	9318677	-
gRNA- Δ t-2	chr2	3123966	-
gRNA- Δ t-2	chr4	54381573	-
gRNA- Δ t-2	chr8	133082998	+
gRNA- Δ t-2	chrY	10978196	-
gRNA- Δ t-2	chr17	39154563	-

Table R1. No potential off-target sites have been found on Chr 10.

The table shows the predicted off-target sites using gRNA- Δ i-1, gRNA- Δ i-2, gRNA- Δ t-1, gRNA- Δ t-2. The prediction was performed using Cas-OFFinder (<http://www.rgenome.net/cas-offinder/>) with the following parameters: PAM type: SpCas9 from *Streptococcus pyogenes*: 5'-NGG-3'; Target Genome: Homo sapiens (GRCh38/hg38) – Human; Mismatch Number: 2; DNA Bulge Size: 1; RNA Bulge Size: 1.

2. The confirmation of the deletion in other HAP1 CRISPR/Cas- clones, occurring in 44-61% of the clones while not being detected in control clones, does not rule out the possibility of a clonal effect. It would be helpful for the authors to investigate whether the deletion occurs in a small population of cells within the bulk HAP1 cell population.

We would like to clarify that we did detect the deletion of *PAPSS2-PTEN* locus among control clones (**Fig. 5B**, first column). These control clones were generated by transfection of the px459 plasmid (into which no gRNA sequence was cloned), followed by puromycin selection and the generation of single cell-derived clone. Based on our experiment depicted in **Fig. 5B**, approximately 67% of the control clones exhibited the deletion of the *PAPSS2-PTEN* locus.

However, it is important to note that the specific clone used for RNA-seq and ChIP-seq analysis, referred to as “ctrl” in **Figs. 1-4**, was experimentally validated to carry an intact *PAPSS2-PTEN* locus.

To address concerns regarding clonal effects and to explore whether we could detect the deletion in bulk HAP1 cells (reviewer 3 referred to those cells as parental “bulk” HAP1 cells), we compared the presence of the *PAPSS2-PTEN* locus between single cell-derived clones and bulk cells. We divided bulk HAP1 cells without any transfection or treatment, into three separate passages and collected their genomic DNA as triplicates at each passage. Additionally, we selected three single cell-derived clones characterized as “with *PAPSS2-PTEN*” (as shown in **Fig. 5B**). To assess the presence of the *PAPSS2-PTEN* locus, we performed qPCR amplifying a genomic region located in the promoter region of *ATAD1* and a genomic region with no annotation located in Chr 12 as the internal control. By comparing the frequencies of the *PAPSS2-PTEN* locus between the single cell-derived clones identified as “with *PAPSS2-PTEN*” and the bulk HAP1 cells, we noted a decrease in the abundance of the *PAPSS2-PTEN* locus with the bulk HAP1 cells. This observation strongly indicates that a fraction of HAP1 cells within the bulk population already possesses the deletion (**Fig. R1**).

Fig R1. Cells without the *PAPSS2-PTEN* locus are present in bulk HAP1 cells.

The barplot shows the relative frequencies of the *PAPSS2-PTEN* locus in genomic DNA molecules extracted from bulk HAP1 cells (left) and single cell-derived clones (right) with the *PAPSS2-PTEN* locus. The frequencies are presented as fold changes, obtained by normalizing the number of molecules carrying a genomic region in the *ATAD1* promoter region to the number of molecules carrying our internal control (n = 3, mean ± SD, p = 0.08).

3. It is unclear whether the unmodified clones and dt-dt1 clones are related. Providing information on the junction sequences at the 10q23.31 deletion and their clonality would be beneficial for better understanding the findings.

To reveal the junction sequences at the 10q23.31 deletion, our initial approach was to perform PCR amplification of the junction followed by Sanger sequencing for sequence identification. We designed two pairs of primers based on the best performance predictions using NCBI Primer-BLAST (https://www.ncbi.nlm.nih.gov/tools/primer-blast/index.cgi?LINK_LOC=BlastHome). However, the PCR amplification yielded nonspecific results with amplicons of varying lengths (**Fig. R2A**). Furthermore, in the case of $\Delta t45$, which retained the *PAPSS2-PTEN* locus, no differences were observed compared to other clones carrying the deletion of the *PAPSS2-PTEN* locus. Based on these findings, we concluded that the amplicons lacked specificity and reliability.

Further analysis of genomic sequences flanking the deletion boundaries in UCSC genome browser revealed a high density of repetitive elements (depicted as the “repeats” track) (**Fig. R2B-D**). Particularly, within an approximately 8 kb region at the 3’ end of the break boundaries, most of the genomic sequences were annotated as repeats (**Fig. R2D**), which tremendously impeded successful PCR amplifications.

To overcome these challenges, we employed ChIP-seq reads to estimate the break boundaries (**Fig. R2E-F**). The position indicated as red line in **Fig. R2E-F** was used as the reference base, expanded to a 300 bp window of 50 bp upstream and 250 bp downstream for 5’ break site (**Fig. R2E**), and 250 bp upstream and 50 bp downstream for 3’ break site (**Fig. R2F**).

For 5' break site:

AAGCAGGTTTCAAATCCAGGTCTGCTTGACTCCAAAATCGATACTCATGC|TACTCCGAT
GCATGCTACAGAGAGTTTTTCTGGGCCAGTAGTGGGAAGAAGGTAGAGATTTTTGCAA
ATTGTCTTAGGGACCCAGTATAGAAGTATGCCACTAACTTTGGGGTTTCTGGACCACTTC
CTGATAAGGGGCAGAGTTAGATGGGATGCCGAGGATGTCTGTGTGTTCTCTATGTACACA
CAGCAAGACAGACATTTTTTCCTTTAACATGTGTTTTGCTTTCTATAGAGAGAATCCTTTT
TC

For 3' breakage site:

TATTCAGTGCTAACAAGGAATAAGCTATCAACTCAGGAAAAGACATGGAAGAACATTAG
ATGCATATGACTGAGTGAAAGAAGCCAATCCAAAAGACTACATACTGTATTACTCCAAC
TATATGACATTTTGA AAAAGGCCAAA ACTATGGAGACAGCTTAAAGATCAGTGGTTGCCA
GGGATTAGGGGTGAGGAAAGGATGAATCGGTAGGGCACAGAGGATTTTCAGGGCAGTA
AAATTATTCAGTATA|ACACTATAATGGTAGATACATCATTACACATTTGTCCAAACCCACA
GCAT

The symbol | denotes the position where the break is presumed to occur. Therefore, in the cells carrying the deletion of *PAPSS2-PTEN* locus, the junction sequence is as follows:

AAGCAGGTTTCAAATCCAGGTCTGCTTGACTCCAAAATCGATACTCATGC|ACACTATAAT
GGTAGATACATCATTACACATTTGTCCAAACCCACAGCAT

Based on this finding, we determined that the deleted region spans 283,142 bp. Consequently, we have made a correction in the manuscript, adjusting the length from “287 kb” to “283 kb”.

Fig R2. The sequences at the breakage boundaries are identifiable by ChIP-seq.

(A) Agarose gels display the sizes of the PCR products obtained by using two sets of primers. Marker bands specify DNA size in bp. The clones carrying the deletion of *PAPSS2-PTEN* locus in Fig. S1B are highlighted in red. (B-D) UCSC Genome Browser views (hg38) show the coverage track of H3K27ac ChIP-seq in $\Delta t1$, the presence of repeats, and gene annotation at the *PAPSS2-PTEN* locus on Chr 10. In the “repeats” track, the genomic regions of single repeats are visualized as black bars. The magnifying view displays the DNA break site boundaries at the (C) 5’ break site and (D) 3’ break site. (E-F) The hg38 genome browser illustrates the reads from one replicate of H3K4me3 and two replicates of H3K27ac ChIP-seq in the $\Delta t1$ clone mapped at the *PAPSS2-PTEN* locus. The red lines represent the positions at which the DNA break is considered to occur. The deleted *PAPSS2-PTEN* locus is colored in red.

4. The authors observed that HAP1 cells show increased resistance to external stress and overall replication stress. It would be informative for the authors to demonstrate whether this phenotype is primarily dependent on PTEN or other genes that are deleted. e.g. overexposes PTEN in some of the HAP1 clones see if it reverts the epigenetic, or gene expression profiles.

We appreciate the reviewer's suggestion. However, restoring PTEN abundance to the same level as in the cells with the *PAPSS2-PTEN* locus is challenging when, for example overexpressing of PTEN by cloning *PTEN* on a vector and delivering this vector into cells. Moreover, direct delivery of the PTEN protein directly to the cell would prevent studying of the regulatory roles of the genomic locus. Therefore, we opted to compare the *PTEN* knockout (KO) cell with the unmodified (wild-type) cells to dissect the impact of the *PTEN* deletion alone on epigenetic or gene expression profiles.

For our analysis of the *PTEN* deletion effect, we utilized two publicly available datasets (GSE61794 and GSE69822) containing RNA-seq data of unmodified cells and *PTEN* KO cells. GSE61794 used mesenchymal stem cell (MSC) and neural stem cell (NSC) as cell models, while GSE69822 employed the breast epithelium cell line (MCF10a). We extracted the differentially expressed (DE) genes from *PTEN* KO cells in MSC, NSC, and MCF10a and compared them to a subset of DE genes in HAP1 cells without the *PAPSS2-PTEN* locus (**Fig. 3**).

We first investigated the overlap between the top 100 downregulated (**Fig. R3A**) or upregulated (**Fig. R3B**) DE genes in Δ *PAPSS2-PTEN* HAP1 cells and those any other *PTEN* KO cells. Approximately 30% of the top 100 deregulated DE genes in Δ *PAPSS2-PTEN* HAP1 cells overlapped with any of the DE genes identified in the three cell types from the published datasets (**Fig. R3A-B**). Next, we examined the DE genes in Δ *PAPSS2-PTEN* HAP1 cells contributing to the top5 enriched KEGG pathways (**Fig. 3F**). Out of the 61 genes, 42 were not deregulated in MSC, NSC or MCF10a *PTEN* KO cells (**Fig. R3C**). Most of the 19 genes downregulated in at least one cell type with the *PTEN* deletion were associated with cell cycle functions (**Fig. R3D**), in line with previous reports linking PTEN in the regulation of cell cycle progression and proliferation (Brandmaier et al. 2017; Song et al. 2012). From these findings, we concluded that the deletion of *PTEN* leads to transcriptional changes in Δ *PAPSS2-PTEN* HAP1 cells, impacting cell behavior as previously reported. However, the deletion of the *PAPSS2-PTEN* locus had a notably greater impact on the transcriptome compared to the *PTEN* deletion alone.

Subsequently, we examined the subset of DE genes in HAP1 cells with proximal H3K27 differentially acetylated regions (≤ 5 kb) and compared them to all the DE genes in MSC, NSC and MCF10a *PTEN* KO cells. Around 62% (71/115) of this DE gene subset in HAP1 did not exhibit differential expression in any of these three cell types (**Fig. R3E**). This suggests that while the *PTEN* deletion contributes to changes in H3K27 differential acetylation, it is not a primary determinant.

Several hypotheses could explain the limited impact of the *PTEN* deletion on the chromatin environment and transcriptomes. Firstly, more additive effects from the *PAPSS2-PTEN* deletion could exist than currently known. Although *PTEN* is considered to be the most critical gene in the $\Delta 10q23$ region, co-deletion of *PTEN* with nearby genes might exert greater influence on cells than *PTEN* gene deletion alone. This idea is discussed in the third paragraph of our discussion section in the manuscript. Lastly, due to the inability to establish the chronological order of the events leading to the *PAPSS2-PTEN* deletion and the chromatin and transcriptome changes, it is plausible that altered H3K27 acetylation and gene expression defects led to excessive DNA replication stress, triggering double-strand DNA breaks at the *PAPSS2-PTEN* locus, as discussed in the fourth paragraph in our discussion section. Therefore, it is not surprising that most molecular changes identified in Δ *PAPSS2-PTEN* HAP1 cells are not directly induced by *PTEN* deletion alone.

Fig R3. Molecular changes identified in Δ PAPSS2-PTEN HAP1 cells are only partly driven by the PTEN deletion.

(A-C, E) Four-way Venn diagrams intersect the numbers of (A) top 100 downregulated and (B) top 100 upregulated DE genes (ordered by fold change) in Δ PAPSS2-PTEN HAP1 cells with any (A) downregulated or (B) upregulated DE genes in MSC, MCF10a and NSC PTEN KO cells, as well as (C) DE genes contributing to the top 5 KEGG enriched pathway (all downregulated, as shown in Fig. 3E-F) in Δ PAPSS2-PTEN HAP1 cells and downregulated DE genes in MSC, MCF10a and NSC PTEN KO cells, as well as (E) DE genes with proximal H3K27 differentially acetylated (DAc) regions (≤ 5 kb) in Δ PAPSS2-PTEN HAP1 cells and deregulated DE genes in MSC, MCF10a and NSC PTEN KO cells. (D) The gene concept network displays genes contributing to the top 5 enriched KEGG pathways (ranked by FDR-adjusted p values) in Δ PAPSS2-PTEN HAP1 cells. Those DE genes identified as differentially expressed in at least one PTEN KO cells in the published datasets (19/61, shown in C) are labelled in red.

Reviewer #2 (Comments to the Authors (Required)):

SUMMARY:

This study reports the inconsistent occurrence of a large and unexpected genomic deletion at 10q23.31 in HAP1 cells, a widely used human haploid cell model. The 287 kb deleted region on chromosome 10 encompasses several protein coding genes that have regulatory functions including PAPSS2, ATAD1, KLLN and PTEN, as well as the noncoding pseudo-gene CFL1P1 (i.e. cofilin-like). The authors report that deletion of this region only occurred in CRISPR/Cas9 edited clones, specifically when they generated variants of two proximal transfer RNA genes on chromosome 17. Deletion of the 10q23.31 region was also found to be associated with global changes in histone acetylation and gene expression, which can alter the internal wiring of a cell and confound biological studies. A key claim in the manuscript is that generation of CRISPR-Cas9 edited cells and/or exposure to cellular stressors greatly increases the occurrence of the 10q23.31 deletion. They also provide evidence that the 10q23.31 deletion is a collateral deletion that occurs frequently in patient tumors.

CRITIQUE:

The observation of 10q23.31 collateral deletion in some clonal HAP1 cell lines is noteworthy, given that HAP1 cells are a commonly used model human cell line. However, mutation and deep deletion at the PTEN locus among patient tumors is quite common and well established. The tendency for co-occurrence of alterations for all combinations of PAPSS2, ATAD1, KLLN and PTEN is highly significant across several different cancer types (cbiportal.org). It is also well known that loss of PTEN can cause abnormal transcript signatures linked to cell cycle and DNA replication phenotypes. Better integration with previously published work reporting the effects of PTEN loss on transcriptome outcomes would elevate the manuscript.

The importance of this study is in the proposed mechanism and frequency by which the Δ PAPSS2-PTEN deletion occurs, which may have broader impacts for engineering cell lines. Obviously collateral PTEN loss could have dire consequences for cell-based therapies. For example, autologous CAR-T based therapies that may incorporate a gene editing step with CRISPR/Cas9 to introduce a beneficial mutation or even the CAR construct itself at a defined locus. To explain the cause of the Δ PAPSS2-PTEN deletion, the authors systematically tested individual steps commonly performed when generating CRISPR/Cas9 deletion clones and ruled out gRNA effects, Cas9 off-targeting, pBluescript co-transfection, transfection of plasmids in general, and ploidy status. The state that "the deletion of the Δ PAPSS2-PTEN locus occurred at low frequency in HAP1 cells devoid of plasmid transfections or antibiotic selection" and "the probability of losing the Δ PAPSS2-PTEN locus significantly increased during the process of generating CRISPR-Cas9 cell clones and can make cells more resilient to cellular stress". Unless I was misunderstanding Figure 5B, it looks like the authors are showing that puromycin selection causes a much higher proportion of Δ PAPSS2-PTEN cells compared with no puromycin treatment. At a minimum, the authors should consider doing the following:

[1] See if the Δ PAPSS2-PTEN deletion occurs with other selection markers (e.g. blasticidin, neomycin, hygromycin, zeocin), and compare the frequency of occurrence.

To investigate whether the Δ PAPSS2-PTEN occurs upon cell exposure to antibiotics commonly used for the selection of CRISPR-Cas9-modified human cell clones, we exposed HAP1 cells at the same confluency to three antibiotics at different concentration: Blasticidine (low: 1 μ g/ml, mid: 5 μ g/ml, high: 10 μ g/ml), G418/Geneticin (low: 200 μ g/ml, mid: 500 μ g/ml, high: 1000 μ g/ml) and Puromycin (low: 0.5 μ g/ml, mid: 1 μ g/ml, high: 2 μ g/ml). As controls, we included HAP1 cells exposed to water and DMSO. No plasmid transfection was performed to facilitate antibiotic resistance. Afterwards, genomic DNA was extracted and subjected to qPCR (Methods) using specific primers, to determine the fold change between *ATAD1* (inside *PAPSS2-PTEN* locus) (*ATAD1_2*, Table S1) relative to two undeleted control regions (internal_ctrl_chr12, Table S1, and primer *GAPDH_1*, Fig. R4). Overall, we observed the absence of *PAPSS2-PTEN* locus upon treatment of HAP1 cells with all three antibiotics, even without single-cell picking. The frequency of Δ PAPSS2-PTEN increased with higher concentrations of G418 (range: 200 μ g/ml- 1000 μ g/ml) and Puromycin (range: 0.5 μ g/ml- 2 μ g/ml). In conclusion, the deletion of the *PAPSS2-PTEN* locus in HAP1 cells can be induced by exposure to cellular stress, such as antibiotic treatment, and this effect may be exacerbated with increasing levels of cellular stress, such as antibiotic treatment followed by single cell clone picking.

Fig R4. Cellular stress induced by antibiotic treatment increases the deletion of the *PAPSS2-PTEN* locus.

Bar plot displays the fold-changes between *PAPSS2-PTEN* gene locus and a control locus of in HAP1 cells (bulk) treated with increasing concentrations of antibiotics (Blasticidine (Blast), Geneticin (G418) and Puromycin (Puro)) as well as water (H₂O) and DMSO (controls). Individual biological replicates are shown (black dots). Black line indicates unchanged genomic alteration at the *PAPSS2-PTEN* locus.

Primer sequence used:

GAPDH_1 fw: GAAGTCAGGTGGAGCGAGG, rev: GGGACACAAGAGGACCTCCA
 ATAD1_2 fw: CAACCTGGGAGAGAACGCTT, rev: GACATCAGGTGCGCGAATTT
 internal_ctrl fw: AGGGACATTGGAATTCTCTACCAT, rev:
 ATGATTTTACCTCCCTCTACTTTCA

[2] If the authors are correct and HAP1 cells are prone to a high frequency of Δ *PAPSS2-PTEN* cells following genome engineering, the authors should use a different technology to demonstrate that this indeed the case (e.g. Zn finger nucleases or TALENs).

Our work describes, for the first time, the observation that the *PAPSS2-PTEN* locus deletion already exists in parental HAP1 cells without having undergone genomic engineering. Additionally, we have demonstrated that the generation of CRISPR-Cas9-modified cell clones, coupled with excessive cellular stress, substantially increases the occurrences of cells displaying this deletion. To validate the significance of our unexpected findings that HAP1 cells carry the unintended *PAPSS2-PTEN* locus deletion, we systematically tested individual steps necessary in generating CRISPR-Cas9 single cell-derived HAP1 cell clones. We considered various factors that could affect the probability of cells with the unintended *PAPSS2-PTEN* locus deletion.

Assessing the nuclease-dependent occurrences of the *PAPSS2-PTEN* locus deletion would be beyond the scope of this study. However, it is important to underscore that our newly discovered insights hold significance because CRISPR-Cas9 genome engineering technology is extensively used in research due to its simplicity and programmability. In contrast, other genome editing methods, such as zinc-finger nucleases (ZFNs) or transcription activator-like effector nucleases (TALENs), require intricate design of tailored DNA binding domains for each target, rendering high-throughput screening impractical. Given that CRISPR-Cas9 technology predominately drives genome genomic engineering and high-throughput screenings, our findings are particularly relevant in this context. The impact on ZFNs- or TALENs-mediated genome editing is comparatively limited.

In conclusion, while we appreciate the reviewer's suggestion, we believe that the systematic testing of various steps of the engineering of CRISPR-Cas9-modified HAP1 cell clones enhances the significance and relevance of our findings. This approach outweighs the option of contrasting different genome editing technologies.

[3] The authors should attempt to report on the frequency of Δ PAPSS2-PTEN deletions in other commonly used cell lines (e.g. RPE1-hTERT, HEK293T, etc..).

We investigated potential *PAPSS2-PTEN* deletions in commonly used cell lines by analyzing 176 ENCODE project histone ChIP-seq profiling datasets from various cell lines on 2023-07-06, including:

A549, 86 datasets;

GM12878, 15 datasets;

HCT116, 17 datasets;

HEK293, 6 datasets;

HepG2, 15 datasets;

K562, 19 datasets,

MCF-7, 18 datasets

Following meticulous manual inspection using the UCSC genome browser (hg38), we have not identified discernable molecular indications pointing towards a genomic deletion at the *PAPSS2-PTEN* locus within the examined cell lines. Importantly, this observation pertains exclusively to the tested datasets obtained from the corresponding cell lines. There is a possibility that other datasets or cell lines, which were not included in our testing due to unavailability of suitable datasets, might indeed harbor this genomic deletion. Therefore, a thorough investigation of the presence of the *PAPSS2-PTEN* locus becomes crucial, especially when cell lines are employed in CRISPR screens.

In addition to these experiments, do the authors know if TP53 status influences the frequency of Δ PAPSS2-PTEN deletion clones in any given population of cells? What would happen in HAP1 cells if you restored them to carry the wild-type allele of TP53?

HAP1 carries a documented *TP53* point mutation, with p53 retaining partial functionality in these cells (Moder et al. 2017; Essletzbichler et al. 2014). We explored the impact of functional p53 or wild-type *TP53* on the frequency of Δ PAPSS2-PTEN cells within a cell population, considering two aspects:

1. p53's influence on HAP1 ploidy and growth inhibition: Previous studies have indicated that p53 influences HAP1 ploidy and restricts cell growth (Olbrich et al. 2017). Although HAP1 cells demonstrate instability of haploidy but stability in the diploid state (Olbrich et al. 2017; Yaguchi et al. 2018), restoring p53 functionality fully would likely increase the proportion of diploid HAP1 cells in a cell population. However, our data showed no discernible correlation between ploidy and the frequency of *PAPSS2-PTEN* locus deletion in HAP1 cells (**Fig. 5C**). As a result, restoring functional p53 is unlikely to impact the fraction of Δ PAPSS2-PTEN cells within the population.
2. p53's role in DNA damage response, genome instability and cell cycle: Restoring fully functional p53 is likely to induce apoptosis or cell cycle arrest in HAP1 cells experiencing DNA double-strand breaks at the *PAPSS2-PTEN* locus. This could potentially lead to a reduced number of cells carrying the deletion.

In conclusion, restoring the wild-type TP53 allele may elevate the number of diploid cells but could concurrently diminish the number of Δ PAPSS2-PTEN cells in a given population.

The authors also state that PAPSS2, ATAD1, KLLN and PTEN "exert essential molecular functions" (line 324) and "Given that this region is important for 3D genome organization and encodes four protein coding genes with essential molecular functions, its unintended deletion may dominate over the expected gRNA-mediated CRISPR-Cas9 genomic deletion and could lead to biological misinterpretations" (lines 335-338). In fact, none of these genes are classified as common or selectively essential genes in the Cancer Dependency Map data so the authors should clarify which specific essential functions they are referring to. Although this reviewer agrees that the

deletion could confound biological interpretation, the authors could also provide some concrete examples of how this could occur.

While we acknowledge that these four protein-coding genes may not strictly meet the defined criteria for being labeled as “essential” according to the Cancer Dependency Map data, it is important to note that mouse knockout studies have demonstrated their significance. For instance, studies involving PTEN homozygous knock-out mice have revealed embryonic lethality and loss of tumor suppression, classifying PTEN as “essential” (Di Cristofano et al. 1998). Within the context of the manuscript, we intended to underscore the significance of these four protein-coding genes. The deletion of *PAPSS2-PTEN* locus has been associated with the disruptions of genes controlling critical biological processes such as DNA replication, DNA damage repair and cell cycle, all of which are considered “essential” for maintaining cellular functions. To prevent any potential confusion arising from readers interpreting “essentiality” based on the Cancer Dependency Map, we have revised the term “essential” to “important” throughout the manuscript.

We thank the reviewer for acknowledging that the potential confounding effect of the deletion on biological interpretations. In addition to the instance highlighted in the manuscript, in which the alteration of the *PAPSS2-PTEN* locus affected differential gene expression and H3K27 acetylation, rather than the intended edited genes in the four datasets (Fig. 3-4), it is crucial to recognize the broader implications, particularly in large-scale CRISPR-Cas9 screens. Our findings indicated that cells lacking the *PAPSS2-PTEN* locus exhibited enhanced survival and growth under stress conditions (Fig. 5E, Fig. S4E). Consequently, during CRISPR-Cas9 screening, the presence of HAP1 cells with the *PAPSS2-PTEN* locus deletion could be enriched. Given that CRISPR-Cas9 screens provide only gRNA sequences to identify a genomic alteration, the distinction between cells with or without the *PAPSS2-PTEN* locus deletion becomes impossible. This predicament further hampers the accurate assessment of whether the deletion itself or the intended gRNA targeting influences the observed functional outcome.

Lastly, the authors show that clones harboring the Δ PAPSS2-PTEN deletion have altered transcriptional profiles, as well as H3K4me3 and H3K27ac profiles. When the authors looked at published RNA-seq datasets where various genes were mutated with CRISPR-Cas9, they did not find any direct interactions with the PAPSS2-PTEN deleted genes, but after performing a principal component analysis, the principal components PC3-PC5 separated samples based on PAPSS2-PTEN presence or absence. It would be interesting to see whether re-expression of 1 or a combination of the deleted genes restores the transcriptional and epigenetic profiles in HAP1 cells. Moreover, starting at line 590 they describe that PAPSS2-PTEN genes are deleted in many cancer cell types, but that PTEN is deleted in the highest number. The deletion of the other genes in this locus (i.e. ATAD1, KLLN and PAPSS2) appears to occur in an accumulative manner following the gene order in the linear genome. Could it be that PTEN drives the majority of the transcriptional and epigenetic profile changes in Cas9-modified HAP1 cells. What would happen if you started the experiment with a PTEN mutant HAP1 cell line? Would this be less prone to Δ PAPSS2-PTEN occurrences? From the clinical data they examine, they find that "deep deletion" occurs commonly in prostate adenocarcinoma cells. It would be informative to see whether CRISPR/Cas9 modification of prostate cancer cell models are particularly sensitive to increased Δ PAPSS2-PTEN deletion rate relative to other cell types. And whether this is true of other cancer types where PTEN deletions are commonly found.

We have divided the comment into three points for clarity:

1) Does re-expression of *PTEN* restore the majority of the transcriptional and epigenetic profile changes in Cas9-modified HAP1 cells?

Re-expressing functional *PTEN* alone does not fully reverse all the observed transcriptional and epigenetic changes. As mentioned in our response to reviewer 1 point 4, re-expressing *PTEN* will only restore a limited number of the observed changes in HAP1 cells.

2) Is *PTEN* mutant HAP1 cell less prone to the deletion?

As we discussed in the response to reviewer 1 point 4, it is possible that the dysregulated genes contribute to excessive DNA replication stress, triggering the deletion of the *PAPSS2-PTEN* locus. If this were the case, there would be no difference in the probabilities of HAP1 cells with the *PAPSS2-PTEN* locus deletion and *PTEN* mutants.

However, if the deletion of the *PAPSS2-PTEN* locus occurs first, *PTEN* mutant HAP1 cells might be less prone to the deletion. Our analysis indicated that the *PTEN* deletion affects the expression of many gene related to cell cycle (**Fig. R3D**), suggesting that HAP1 cells with mutated *PTEN* may partially resemble the growth pattern of Δ *PAPSS2-PTEN* HAP1 cells.

3) Does CRISPR-Cas9 modified prostate cancer cell models are more prone to Δ *PAPSS2-PTEN* deletion? What about other cancer types where *PTEN* is frequently deleted?

As noted by the reviewer stated, collateral deletions at the *PTEN* locus are common in prostate cancer. Many commonly used prostate cancer cell lines, originating from prostate cancer patient, carry a *PTEN* deletion. For example, PC-3 cells do not express *PTEN* due to the homozygous *PTEN* gene deletion (Schmitz et al. 2007), and LNCaP cells carry one allele with the *PTEN* deletion and one allele with a *PTEN* mutation (Lotan et al. 2011). This suggests *PTEN* locus deletions in prostate cancer cells. However, evaluating CRISPR-Cas9 gene editing events in prostate cancer cells in which *PTEN* is already deleted may have limited relevance. Collateral deletion at *PAPSS2-PTEN* frequently are also frequent in glioblastoma, with the widely used glioma cell lines H4 harboring the 10q23 deletion (Meléndez et al. 2011).

Reviewer #3 (Comments to the Authors (Required)):

The paper by Geng et al., describes an interesting finding that CRISPR-Cas9-modified HAP1 cells obtained an unexpected deletion of 10q23.31 region that contains four recurrently deleted genes in human cancers, including the tumor suppressor gene *PTEN*. The authors initially observed 10q23.31 loss from chromatin profiling data in CRISPR-Cas9-modified HAP1 cells, and further investigated 10q23.31 loss in a panel of diverse human cancer patient cells using public databases. The authors found that 10q23.31 loss caused significant changes in both H3K27ac occupancy and transcription level of a number of cell cycle and DNA replication related genes. The altered transcription signature was also validated using public human cancer patient databases. In an effort to elucidate the mechanism involved in generating 10q23.31 loss in CRISPR-Cas9-modified HAP1 cells, the authors concluded that the process of CRISPR-Cas9-mediated gene editing triggered 10q23.31 loss, and genotoxic stress potentially enhanced the frequency of 10q23.31 loss. In line with this, radiotherapy induced more 10q23.31 loss in clinical. Although the mechanism of 10q23.31 loss in cancer patient cells is still elusive, this paper carefully explored and discussed multiple potential factors during CRISPR-Cas9-mediated gene targeting that could affect the frequency of 10q23.31 deletion.

Overall, this manuscript demonstrated an important recurrent phenomenon caused by CRISPR-Cas9-mediated gene editing in HAP1 cells, which may represent a general caveat of using CRISPR-Cas9-mediated gene editing in other cell lines as well and reflect the similar genetic event that occurs in diverse cancer patient cells. The authors provided some mechanistic and clinical insights on the recurrent 10q23.31 loss in human cancers. There is already a lot of supportive data for this discovery. However, there are still some concerns when addressed would further strengthen this manuscript.

1, Whether the dual gRNA system targeting two proximal transfer RNA (tRNA) genes caused off-target effect to generate Δ t1 and Δ i17 clones?

We utilized Cas-OFFinder to predict potential off-target sites for our four gRNAs used to generate Δ t and Δ i clones (**Table R1**). Please find the details (**Table R1**) in the response to Review 1, Point

1. Our prediction analysis suggested that off-target effects are minimal since, primarily because the majority of predicted off-target sites require two mismatches and one bulge simultaneously, which is typically poorly tolerated by the CRISPR-Cas9 system.

Importantly, we excluded the possibility that the deletion of *PAPSS2-PTEN* locus results from off-target effect related to our dual gRNA system targeting proximal transfer RNA (tRNA) genes. This conclusion is based on the following evidence:

1. We found the deletion of the *PAPSS2-PTEN* locus in publicly available data, in which protein-coding genes (*METAP1*, *SREBF2* and *SMARCC1*) were targeted in the HAP1 genome using very different sgRNA sequences from those of ours used to generate tRNA deletions (Fig. 2A-C).
2. We identified that the deletion of the *PAPSS2-PTEN* loci occurs in HAP1 cell clones even without delivering CRISPR-Cas9 components to the cells, confirming that the deletion is not a result by Cas9 off-target cleavage (Fig. 5B).

Did the author use two control sgRNA pairs for the CRISPR-Cas9-unmodified control (ctrl) clone? In our control clones, we used a non-targeting sgRNA. For the CRISPR-Cas9 unmodified (ctrl) clone, we delivered the px459 vector into the cells without cloning any gRNA sequence. This was followed by puromycin selection and the generation of single cell-derived clones.

Although no specific gRNA sequence was cloned, there are nucleotides in the backbone of the px459 vector positioned between the human U6 promoter and the chimeric guide RNA scaffold (Fig. R5). This short sequence would have been excised through restriction enzyme digestion if gRNA sequences were cloned. However, as we directly transfected the cells with the px459 vector, the short sequence was retained and Pol III transcribed, effectively serving as a control gRNA in HAP1 cells.

It is important to note that this short sequence does not exhibit a perfect match to the human reference genome (hg38). The best alignment in the human genome corresponds to a genomic region on chromosome 12 (Chr12:123,148,517-123,148,502). Nevertheless, there are mismatches at the 5' and the 3' end of this sequence. It is well-known that the CRISPR-Cas9 system is intolerant towards mismatches at the 3' end (the nucleotides next to the PAM sequence, 5'-NGG-3'). Notably, no PAM sequence is found within this sequence. Therefore, this short sequence in the px459 vector serves as a control gRNA in the generation of our ctrl clones.

Fig R5. A short sequence in the px459 vector served as the control gRNA.

SnapGene Viewer visualizing the px459 vector shows the short sequence (indicated by red square) between the human U6 promoter and the chimeric guide RNA scaffold.

2, The authors made conclusion that "Since the guide RNA sequences (gRNAs) used for generating the Δt and Δi cell clones were different, we concluded that the 10q23.31 deletion occurred in a gRNA-independent manner". The fact that "the guide RNA sequences (gRNAs) used for generating the Δt and Δi cell clones were different" is supportive for conclusion that 10q23.31 deletion occurred in a gRNA sequence-independent manner. If so, the author should be clear about what control gRNAs they used, and the clone with control gRNA pairs should also have the same deletion; in contrast, the parental cells would have no 10q23.31 deletion.

The authors should also consider whether gRNA-guided genetic cutting by CRISPR-Cas9 is

required for triggering 10q23.31 loss, or transfection of CRISPR-Cas9 is sufficient to cause 10q23.31 deletion?

We apologize for any confusion and would like to clarify our findings. We state that the 10q23.31 deletion occurred in a gRNA-independent manner when describing the *PAPSS2-PTEN* locus deletion in $\Delta t1$ and $\Delta i17$ clones. Our subsequent investigation revealed that certain HAP1 cell clones had lost the *PAPSS2-PTEN* loci without any CRISPR-Cas9 components being delivered to the cells (Fig. 5B, last 4 groups in the barplot). This observation suggests that gRNA sequences and the CRISPR-Cas9 system, in general, are not required for triggering the 10q23.31 deletion, but the use of the CRISPR-Cas9 system greatly increased the frequency of this deletion. Moreover, even cell clones that had not undergone any transfection or puromycin selection carried the *PAPSS2-PTEN* locus deletion (Fig. 5B, last group in the barplot). Our analysis of DNA molecules, assessing the frequency of the *PAPSS2-PTEN* locus in both bulk parental HAP1 cells and in single cell-derived clones, further confirmed the presence of the deletion in parental bulk HAP1 cells.

Regarding the control clones, please refer to the details in the response to Reviewer 3, point 1. It is important to note that some of our control clones do indeed carry the *PAPSS2-PTEN* locus deletion, underscoring the presence of the deletion in the parental HAP1 cells. While the control clone (ctrl) used in the manuscript (Fig. 1-4) retained the *PAPSS2-PTEN* locus, another control clone (c3) carried a *PAPSS2-PTEN* locus deletion (Fig. R6B).

To conclude, we have identified control clones containing this deletion, and characterization of one of them (c3) highlighted that the deletion neither arises from Cas9 cleavage nor is dependent on the presence of CRISPR-Cas9 components. However, our data demonstrate that applying CRISPR-Cas9 genome editing to the cells increases the likelihood of cells losing the *PAPSS2-PTEN* locus.

Fig R6. The control clone c3 carries the *PAPSS2-PTEN* locus deletion.

(A-B) The hg38 genome browser view shows normalized RNA-seq coverage tracks for the plus (+) and minus (-) strand as well as H3K27ac ChIP-seq reads in the HAP1 control clone c3 at (A) the targeted tRNA genes loci and (B) the *PAPSS2-PTEN* locus (highlighted by a red box).

3, The authors found that "Hi-C and ChIA-PET data obtained from HAP1 and other cell types revealed multiple long-range interactions between the 10q23.31 and other genomic regions indicating additional roles in three-dimensional (3D) gene regulation". One wonders whether those gRNAs that triggered 10q23.31 deletion targeted the regions involved in 10q23.31's long-range interactions, thereby, facilitated 10q23.31 deletion during the process of CRISPR-Cas9-induced gene editing/repair?

In the manuscript, we mentioned that several genomic loci located on Chr 10 interact with the *PAPSS2-PTEN* locus. However, our off-target prediction for the four gRNAs used did not reveal

any potential off-target cleavage at 10q23.31 (**Table R1**). Please find more details in our response to reviewer 1, Point 1. Therefore, it is unlikely that an off-target cleavage at the region involved in the long-range interactions at 10q23.31 facilitated the *PAPSS2-PTEN* locus deletion. Moreover, our data indicated that puromycin selection of cells without antibiotic resistance greatly increased the frequency of the *PAPSS2-PTEN* locus deletion in the absence of any CRISPR/Cas9 components (**Fig. 5B**).

4, The authors showed the chromatin profiling of H3K4me3 and H3K27ac to visualize 10q23.31 deletion. How did Input or IgG control chromatin profiling look like at the same regions?

To address this question, we generated bedgraph to visualize the following data (**Fig. R7**):

1. Input for our HAP1 control clone ctrl (with an intact *PAPSS2-PTEN* locus) and the CRISPR-Cas9-modified clone $\Delta t1$ (with Δ *PAPSS2-PTEN*), as used in **Fig. 1-4** and **Fig. S1-3**.
2. IgG profiles for HAP1 clone in the published dataset 3: Δ SM4 replicate 1 (Δ SM4_1, with *PAPSS2-PTEN*), Δ SM4 replicate 2 (Δ SM4_2, with *PAPSS2-PTEN*), Δ SM1 replicate 1 (Δ SM1_1, Δ *PAPSS2-PTEN*) and Δ SM1 replicate 2 (Δ SM1_2, Δ *PAPSS2-PTEN*), as used in **Fig. 4** and **Fig. S3**.

In all the input and IgG control from the HAP1 clones with a *PAPSS2-PTEN* locus deletion, we detected an aberrant read distribution, suggestive of a genomic deletion at the *PAPSS2-PTEN* locus. This observation aligns with the findings in H3K4me3 and H3K27ac ChIP-seq data (**Fig. R7**).

Fig R7. Signal depletion of input or IgG chromatin profiles at the *PAPSS2-PTEN* locus in Δ *PAPSS2-PTEN* HAP1 clones.

The hg38 genome browser view shows reads obtained from input or IgG chromatin profiling in the HAP1 control clone (ctrl) 3 $\Delta t1$, Δ SM4_1, Δ SM4_2, Δ SM1_1 and Δ SM1_2 at the *PAPSS2-PTEN* locus (highlighted by a red box). Read color code differentiates HAP1 clones without (Δ *PAPSS2-PTEN*, orange) and with the *PAPSS2-PTEN* locus (grey).

5, "We next examined the top 20 genes contributing to PC1 and PC2. ATAD1, PTEN and PAPSS2 were among the top genes that contributed to the strongest separation of the two groups (Fig. 2D). None of the top genes included the originally intended gene knockouts (Fig. 2D) and were located on different chromosomes (Supplementary Table S4-5)." This part is confusing. ATAD1, PTEN and PAPSS2 were among the top 20 genes and were located on the same chromosome, why the authors claimed that "None of the top genes were located on different chromosomes". In addition, before batch correction, did authors see predominant expression changes induced by intended gene editing? What kind of controls (parental or control gRNAs) the other three datasets used?

We appreciate the reviewer for pointing out the confusion in our previous description. We have revised the text to read as follows: "None of the top genes included the originally intended gene knockouts (**Fig. 2D**). Besides *ATAD1*, *PTEN* and *PAPSS2*, the other top 20 genes were located on different chromosomes (**Supplementary Table S4-5**)."

Before batch correction, PC1 (52%) and PC2 (29%) distinctly separated the samples by datasets. All top 20 genes contributing to PC1 and PC2 drove the separation of samples from dataset 1 compared to the rest of the samples (**Fig. R8A**). Most of these top 20 genes contributing to PC1

and PC2 are non-coding RNA genes such as *RN7SL4P*, *RN7SK*, *SNORD3D* and *SNORD3A*. We subsequently plotted the top 200 genes contributing to PC1 and PC2. Out of these 200 genes, 198 drove the separation of the samples from dataset 1 and dataset 3 (**Fig. R8B**). Notably, none of the originally intended edited genes including *METAP1*, *SREBF2*, *C12orf49*, *SMARCC1* and *SMARCC2* were among these top 200 genes contributing to PC1 and PC2.

The substantial variances observed between samples from dataset 1 and dataset 3, compared to the rest of the samples, could be explained by the sequencing methods employed. In dataset 1 and dataset 3, single-end sequencing instead of pair-end sequencing was used, which likely introduced biases when mapping multi-copy genes families. This could also account for the abundance of non-coding genes from multi-copy gene families among the top ranked genes contributing to PC1 and PC2 separation. Therefore, we concluded that, prior to batch correction, there were no predominant expression changes induced by the intended gene editing.

Importantly, in the studies providing these three published datasets, wild-type/parental HAP1 cells were used.

Fig R8. The intended gene editing did not drive predominant expression changes.

(A-B) Factorial map of the PCA after batch effect correction is shown for the top (A) 20 and (B) 200 genes (purple arrows) separating the HAP1 cell clones that contain (with, grey circle) or lost (Δ , orange circle) the *PAPS2-PTEN* locus in our and other datasets (geometric shapes) in PC1 and PC2.

6, The authors used published RNA-seq datasets of CRISPR-Cas9 edited HAP1 cells from other separate studies. Are these datasets generated from single population clones as well? While the authors used dual gRNA system, one wonders whether the other studies used single gRNA system? If so, what is the authors' thoughts on the effect of different CRISPR-Cas9 methods on the frequency of 10q23.31 deletion. How frequent and how long it takes to generate 10q23.31 deletion upon CRISPR-Cas9 gene editing? And how much this can reflect 10q23.31 deletion in naturally occurred cancer patient cells?

The modified HAP1 cells used in the published RNA-seq datasets were commercially obtained, and no information on single/dual gRNA systems was reported. However, we retrieved information regarding to the molecular signatures of the targeted gene:

- Dataset 1: *METAP1* KO, 179 bp insertion.
- Dataset 2: *SREBF2* KO, 2 bp deletion.
- Dataset 3: *SMARCC1* KO, 4 bp deletion.

Based on these signatures, we speculate that the clones lacking the *PAPS2-PTEN* locus were generated with the single gRNA system.

Our data presented in **Fig. 5** indicate that the deletion was already present in the parental HAP1 cell population used to initiate the CRISPR-Cas9 editing and independent of the use of either single or dual guide RNAs. It appears that excessive cellular stress (e.g., induced by either applying CRISPR-Cas9 to the cells or puromycin selection in the absence of resistant genes) can greatly increases the frequency of the deletion. Compared to the single gRNA system, the dual gRNA system may lead to a slightly higher frequency of the deletion since it induces two on-target double-strand breaks, thereby triggering a higher DNA damage response.

Regarding the reviewer's question about the timeline for generating the deletion, we would like to clarify that it is challenging to determine whether the higher fraction of cells losing the

PAPSS2-PTEN locus after the CRISPR-Cas9 application results from cells with the *PAPSS2-PTEN* locus gaining a growth advantage or the *de novo* generation of deletion. According to our data, the deletion can be observed as early as the second passage after the delivery of CRISPR-Cas9 components.

We hypothesize that the deletion in HAP1 cells and the collateral deletion pattern in cancer patients are both linked to excessive stress. For example, DNA replication stress could be a contributing factor leading to both the *PAPSS2-PTEN* locus deletion in HAP1 cells and in cancer patients. From this perspective, the deletion in HAP1 cells may reflect the collateral deletion observed in cancer patient. For more details, please refer to the discussion section of the manuscript.

7, Whether these differentially expressed genes and H3K27ac changes caused by loss of *PAPSS2-PTEN* locus is not convincing. Would knockout of *PAPSS2-PTEN* in control cells or restoration of *PAPSS2-PTEN* in Δ *PAPSS2-PTEN* cells mimic or rescue the transcription and H3K27ac changes? This point was partly addressed in our response to reviewer 1 point 4 and reviewer 2's last question. As suggested by the reviewer 3, we investigated whether the differentially expressed (DE) genes and H3K27 differentially acetylated regions identified in **Fig. 3-4** in our study also exhibit similar changes when comparing control cells with and without the *PAPSS2-PTEN* locus.

For this analysis, we compared two control clones for which we obtained RNA sequencing data: "c3" without the *PAPSS2-PTEN* locus (shown in response above, **Fig. R6**) and "ctrl" with the *PAPSS2-PTEN* locus used in **Fig. 1-4**. First, we examined the expression levels of DE genes identified in **Fig. 3** (where we compared HAP1 Δ t clones with and without the *PAPSS2-PTEN* locus) in HAP1 control clones c3 (1 replicate, without *PAPSS2-PTEN* locus) and ctrl (2 replicates, with the *PAPSS2-PTEN* locus). We found that 67% of the DE genes from **Fig. 3** exhibited the same trends of gene expression changes between the control clones c3 and ctrl (**Fig. R9A**, left, highlighted by the boxes). These expression trends closely align despite batch biases and normalization differences between the comparisons in **Fig. 3** and between control clones c3 and ctrl. In particular, cluster 1 genes showed higher expression in c3 when compared to ctrl (**Fig. R9A, left**), mirroring upregulation in HAP1 Δ t clones without and with the *PAPSS2-PTEN* locus, respectively (**Fig. 3**). Conversely, cluster 2 and cluster 3 genes exhibited lower expression levels in c3 than ctrl, were also mostly downregulated in HAP1 cell clones without the *PAPSS2-PTEN* locus deletion compared to clones with the intact the *PAPSS2-PTEN* locus (**Fig. 3**). Merging of the two replicates of ctrl (**Fig. R9A, right**) maintained consistent DE gene trends between c3 and ctrl, aligning with gene expression changes between HAP1 Δ t clones without and with the *PAPSS2-PTEN* locus (**Fig. 3**).

Second, we compared the enrichment profiles of c3 and ctrl at differentially acetylated (DAc) regions identified in **Fig. 4** and **Fig. S3**, revealing both gains (**Fig. R9B, left**) and losses (**Fig. R9B, right**) of H3K27ac in c3 compared to ctrl.

In conclusion, our analysis comparing the transcriptomes and H3K27ac profile in c3 and ctrl clones confirms that the DE genes and DAc regions identified in our study (**Fig. 3-4**) are linked to the absence of the *PAPSS2-PTEN* locus.

Lastly, we would like to emphasize that the DE genes and DAc regions are associated with the deletion of *PAPSS2-PTEN* locus, as we stated in the manuscript. We did not assert that the DE genes and DAc regions were solely caused by the deletion because we cannot establish the chronological order of the molecular changes, including the occurrence of the deletion, epigenetic and transcriptomic alterations. It is possible that altered H3K27ac and the disruption in the expression of certain genes preceded the excessive DNA replication stress, ultimately leading to the induction of double-strand breaks at the *PAPSS2-PTEN* locus.

Fig R9. Differentially expressed genes and acetylated regions were dysregulated in the control clone carrying *PAPSS2-PTEN* locus deletion.

(A) Heatmaps display the expression levels of DE genes (identified in the manuscript **Fig.3**) in HAP1 clone ctrl and c3. The data were row-wise scaled based on TPM. Color codes: z-score > 0 (red) and z-score < 0 (blue); row annotation for DE genes: downregulated (aqua) and upregulated (pink); column annotation: with *PAPSS2-PTEN* (ctrl, grey) and Δ *PAPSS2-PTEN* (c3, orange). In the left panel, the two replicates of the ctrl clone were plotted individually, and cluster 1, 2, 3 are indicated with black square with the number in the front. In the right panel, the average of the two ctrl replicated is displayed. (B) Profile plots illustrate the enrichment of H3K27ac in ctrl and c3 clones at the DAC regions (identified in the manuscript **Fig.4** and **Fig.S3**), separated by gain (left) and loss (right).

8, "This result verified that the genomic deletion of the *PAPSS2-PTEN* locus was gRNA-independent and likely caused during the generation of CRISPR-Cas9 deletion clones." This is confusing. While the observation that multiple different gRNA with Cas9 triggered *PAPSS2-PTEN* loss supported that deletion of the *PAPSS2-PTEN* locus was gRNA sequence-independent, whether or not gRNA-mediated Cas9 cutting was required for *PAPSS2-PTEN* loss is unclear. The author compared Cas9 without or with gRNA and did not see significant difference in the frequency of *PAPSS2-PTEN* loss, indicating that gRNA-mediated cutting was not required. This is inconsistent with what the author claimed in the referred statement that "caused by generation of deletion".

We apologized for the confusion. The following amendments have been made in the manuscript to clarify the independency of gRNA. Text changes are highlight in bold:

- "we concluded that the 10q23.31 deletion occurred in a gRNA **sequence-independent manner**".

- “Thus, our inspection of transcriptome data from various CRISPR-Cas9-modified HAP1 cell clones confirmed a gRNA **sequence-independent** deletion at the *PAPSS2-PTEN* locus.”
- “To **study whether gRNA-mediated Cas9 cutting in a sequence-independent manner was required for the deletion of *PAPSS2-PTEN* locus**, we transfected...”
- “This result verified that the genomic deletion of the *PAPSS2-PTEN* locus is **independent of the gRNA sequence and presence** and likely caused during the generation of **CRISPR-Cas9 clones and non-targeting control clones**.”
- To clarify “the generation of CRISPR-Cas9 deletion clones”, we have changed the text from “generation of CRISPR-Cas9 deletion clone” to “**the generation of CRISPR-Cas9 deletion clones and non-targeted control clones**”.

9, The authors should mention in line 541-544 that puromycin selection was applied to the culture although the cells did not receive puromycin resistance gene. This is different from the next paragraph that puromycin selection was omitted. The number of single cell clones used for analysis varied a lot from 50 or so to 3. One worried that statistics from those small amounts of samples may cause significant bias to draw conclusion. In addition, one wonders the efficiency of these transfection assays, especially without selection marker, how the authors distinguish transfected versus non-transfected cells and how this would affect the conclusion?

We have revised the manuscript to make it clear that puromycin selection has been applied. Text changes are highlight in bold:

- “we assessed the occurrence of the *PAPSS2-PTEN* deletion in HAP1 cell clones that were transfected with or without the small-size plasmid (pBlueScript) that did not encode a puromycin resistance gene **and were exposed to puromycin selection**.”

Regarding to the reviewer’s concerns about the number of single cell clones used for analysis, we acknowledge that the number of clones used in **Fig. 5B** varied from 18 clones to 3 clones. In the group with 3 clones, cells were transfected with pBlueScript and exposed to puromycin. We did not draw any conclusions regarding to the statistical significance when comparing this group with 3 clones to other groups with more clones. We only included these 3 clones to demonstrate that even without any CRISPR-Cas9 component, some HAP1 clone still carried *PAPSS2-PTEN* deletion.

We also agree with the reviewer that without selection, it is very difficult to sperate cells with plasmids from those without plasmids after transfection. HAP1 is known as a hard-to-transfect human cell lines (Alexander et al. 2019), and its low transfection efficiency underscores the necessity of selection. Our data highlight the significance of our assessment regarding the influence of puromycin selection on the occurrence of the deletion.

Additionally, although cells with and without plasmids co-existed after transfection when the puromycin selection was omitted, our data showed that exposing cell to transfection reagent and DNA complex did not exacerbate the deletion of *PAPSS2-PTEN* locus (**Fig. 5B**).

10, Transfection without antibiotic selection had reduced occurrences of PAPSS2-PTEN loss may be due to non-transfected parental cells. Transfection with Cas9-puromycin or puromycin alone with antibiotic selection could determine whether Cas9 itself affects the frequency of PAPSS2-PTEN loss. Why transfection with or without plasmids still can induce PAPSS2-PTEN loss, albeit much lower frequency?

Our data indicated that in the bulk parental HAP1 population, a small fraction of cells carried the deletion of *PAPSS2-PTEN* locus. Consequently, HAP1 clones with the *PAPSS2-PTEN* locus deletion were still found, even without transfection or puromycin selection. We subsequently confirmed the presence of cells carrying the deletion in bulk parental HAP1 cells. For more details, we refer the reviewer to our response to reviewer 1, point 2.

11, Did the authors also see increased frequency of PAPSS2-PTEN loss in drug treated patients or cells? Is PAPSS2-PTEN loss more related to genotoxic stress/exposure compared with non-

genotoxic agents?

Yes, we did observe an increased frequency of *PAPSS2-PTEN* locus deletion when HAP1 cells were treated with Puromycin and other drugs, such as Blasticidine and Geneticin (G418) (**Fig. 5B, Fig. R4**). Unfortunately, we did not find information regarding to the drug treatment of the published patient datasets used in this study.

We only tested the drugs interfering with protein synthesis (**Fig. 5B, Fig. R4**). However, upon considering genotoxic stress, it is likely that a higher frequency of cell lose the *PAPSS2-PTEN* locus when exposed to genotoxic agents than non-genotoxic agents.

12, Did the authors observe correlation between oncogene overexpression (eg. Myc) and *PAPSS2-PTEN* loss in cancer patient samples?

We selected five well-known oncogenes: *ERBB2*, *KRAS*, *MYC*, *MAPK1* and *MAPK3* and plotted their expression levels in cancer patient samples that were chosen in **Fig. 6C-E** and **Fig. S5 (Fig. R10)**. No overexpression pattern or any correlation between oncogenes and the loss of *PAPSS2-PTEN* locus was observed in prostate adenocarcinoma, lung squamous cell carcinoma, skin cutaneous melanoma, and uterine corpus endometrial carcinoma.

Fig R10. The expression of oncogenes was not correlated with the loss of *PAPSS2-PTEN* locus. Heatmaps display the expression levels of selected oncogenes in prostate adenocarcinoma, lung squamous cell carcinoma, skin cutaneous melanoma and uterine corpus endometrial carcinoma. The data were \log_{10} transformed. Color codes: z-score > 0 (red) and z-score < 0 (blue) Patient samples on the top of each heatmap are colored by groups with *PAPSS2-PTEN* (grey) and Δ *PAPSS2-PTEN* (orange).

13, some minor comments:

The authors may have mistakes in lines 370-375 regarding dataset 3 of Δ SM1 or Δ SM2. The term gRNA-independent is unclear. It could mean gRNA sequence-independent, but gRNA is required for Cas9-mediated gene editing to induce *PAPSS2-PTEN* loss, or the authors meant Cas9-mediated cutting without gRNA. The authors may clarify this. Some typos, such as in line 668, 700, etc.

We thank the reviewer for pointing them out for us. We have revised the manuscript accordingly.

Reference

- Alexander J, Findlay GM, Kircher M, Shendure J. 2019. Concurrent genome and epigenome editing by CRISPR-mediated sequence replacement. *BMC Biol* **17**. /pmc/articles/PMC6862751/ (Accessed July 8, 2023).
- Bae S, Park J, Kim JS. 2014. Cas-OFFinder: a fast and versatile algorithm that searches for potential off-target sites of Cas9 RNA-guided endonucleases. *Bioinformatics* **30**: 1473–1475. <https://dx.doi.org/10.1093/bioinformatics/btu048> (Accessed July 8, 2023).
- Brandmaier A, Hou SQ, Shen WH. 2017. Cell cycle control by PTEN. *J Mol Biol* **429**: 2265.

- Essletzbichler P, Konopka T, Santoro F, Chen D, Gapp B V., Kralovics R, Brummelkamp TR, Nijman SMB, Bürckstümmer T. 2014. Megabase-scale deletion using CRISPR/Cas9 to generate a fully haploid human cell line. *Genome Res* **24**: 2059.
- Di Cristofano B, Pesce C, Cordon-Cardo P, P Pandolfi. 1998. Pten is essential for embryonic development and tumour suppression. *Nat Genet* **4**: 348-55.
- Lotan TL, Gurel B, Sutcliffe S, Esopi D, Liu W, Xu J, Hicks JL, Park BH, Humphreys E, Partin AW, et al. 2011. PTEN Protein Loss by Immunostaining: Analytic Validation and Prognostic Indicator for a High Risk Surgical Cohort of Prostate Cancer Patients. *Clin Cancer Res* **17**: 6563.
- Meléndez B, García-Claver A, Ruano Y, Campos-Martín Y, Lope AR de, Pérez-Magán E, Mur P, Torres S, Lorente M, Velasco G, et al. 2011. Copy Number Alterations in Glioma Cell Lines. *Glioma - Explor Its Biol Pract Relev*.
- Moder M, Velimezi G, Owusu M, Mazouzi A, Wiedner M, Ferreira Da Silva J, Robinson-Garcia L, Schischlik F, Slavkovsky R, Kralovics R, et al. 2017. Parallel genome-wide screens identify synthetic viable interactions between the BLM helicase complex and Fanconi anemia. *Nat Commun* **2017 81 8**: 1–8.
- Olbrich T, Mayor-Ruiz C, Vega-Sendino M, Gomez C, Ortega S, Ruiz S, Fernandez-Capetillo O. 2017. A p53-dependent response limits the viability of mammalian haploid cells. *Proc Natl Acad Sci U S A* **114**: 9367–9372.
- Schmitz M, Grignard G, Margue C, Dippel W, Capesius C, Mossong J, Nathan M, Giacchi S, Scheiden R, Kieffer N. 2007. Complete loss of PTEN expression as a possible early prognostic marker for prostate cancer metastasis. *Int J Cancer* **120**: 1284–1292.
- Song MS, Salmena L, Pandolfi PP. 2012. The functions and regulation of the PTEN tumour suppressor. *Nat Rev Mol Cell Biol* **2012 135 13**: 283–296.
- Yaguchi K, Yamamoto T, Matsui R, Tsukada Y, Shibamura A, Kamimura K, Koda T, Uehara R. 2018. Uncoordinated centrosome cycle underlies the instability of non-diploid somatic cells in mammals. *J Cell Biol* **217**: 2463–2483.

October 20, 2023

RE: Life Science Alliance Manuscript #LSA-2023-02128-TR

Dr. Claudia Kutter
Karolinska Institutet
Microbiology, Tumor and Cell Biology, Science for Life Laboratory
Solnavagen 9
Stockholm 171 77
Sweden

Dear Dr. Kutter,

Thank you for submitting your revised manuscript entitled "Intrinsic 10q23.31 deletion is aggravated on CRISPR-mediated engineering mimicking cancer profiles". We would be happy to publish your paper in Life Science Alliance pending final revisions necessary to meet our formatting guidelines.

- please address the remaining minor Reviewer comments
- please note that titles in the system and on the manuscript file must match
- please add your main, supplementary figure, and table legends to the main manuscript text after the references section
- all uploaded datasets should be made publicly accessible at this time, to remove the need for Reviewer passwords. Please update the Data Availability statement accordingly.

A. FINAL FILES:

B. MANUSCRIPT ORGANIZATION AND FORMATTING:

**Submission of a paper that does not conform to Life Science Alliance guidelines will delay the acceptance of your

manuscript.**

The license to publish form must be signed before your manuscript can be sent to production. A link to the electronic license to publish form will be sent to the corresponding author only. Please take a moment to check your funder requirements.

Sincerely,

Reviewer #1 (Comments to the Authors (Required)):

In their research article, Geng et al. made an unexpected discovery regarding a 287 kb region on Chromosome 10 (10q23.31) in chronic myelogenous leukemia HAP1 cells, commonly used in CRISPR/Cas9 screens. They found a deletion of this region consisting of four protein coding genes: PAPSS2, ATAD1, KLLN, and PTEN. Interestingly, the deletion was not attributed to CRISPR/Cas enzymatic activity but was rather generated or promoted during the CRISPR-Cas engineering process itself. The deletion had significant consequences, as it led to global changes in histone acetylation and gene expression, affecting vital cellular processes like the cell cycle and DNA replication.

The reviewer previously suggested that there is also the possibility that the deletion occurred in a subset of HAP1 cells. The reviewers suggested that the authors should discuss the possibility of clonal evolution or provide evidence for the presence or absence of the clonal deletion in HAP1 cells. Additionally, the authors should investigate the presence of off-target sites during sgRNA design or provide evidence that no homologous sequences are found at the Chr10 deletion boundaries. Finally, it is unclear whether the unmodified clones and dt-dt1 clones are related. Providing information on the junction sequences at the 10q23.31 deletion and their clonality would be beneficial for better understanding the findings.

The reviewer's major and minor comments were fully addressed by the authors. First, the authors discussed the clonal evolution effect in the discussion section. Second, the authors demonstrated that the deletion at Chr10 was not derived from CRISPR/Cas off-target activity. Third, the authors demonstrated that the PAPSS2-PTEN locus was already lost in a considerable number of cells/alleles in the "bulk" HAP1 population (Fig. R1). Lastly, the authors concluded the junction for the dt1 clone with the ChIP-seq reads, and the authors acknowledged that the breakpoint was not defined by split reads, as shown in Figure R2, and concluded that this is a presumed location for the dt1 clone.

Minor comments:

The reviewer suggested that the authors include Figure R1 and Figure R2 in the revised manuscript to support their discussion of clonal evolution, but the decision is up to the authors. Otherwise, the reviewer is generally satisfied with the revised manuscript.

Reviewer #2 (Comments to the Authors (Required)):

The response to point 1 concludes that the deletion of the PAPSS2-PTEN locus in HAP1 cells can be induced by exposure to cellular stress (i.e. treatment with different selection antibiotics such as blasticidin, G418 or puromycin), in a dose-dependent manner nonetheless. These are fairly specific cellular stressors, so this should be considered in the messaging in the manuscript, and may provide much more insight into the mechanism behind the PAPSS2-PTEN if contrasted to other cellular stressors.

It was disappointing to see that the authors did not address point 2 and demonstrate that the PAPSS2-PTEN locus deletion could also be induced by non-CRISPR/Cas9 genome editing technologies such as ZnF or TALEN approaches, for which there are many reagents readily available for targeting various human loci. My expectation is that the genome engineering strategy at play for aggravating the 10q23.31 deletion is probably irrelevant, and the authors have certainly not ruled this out.

Reviewer #3 (Comments to the Authors (Required)):

The paper by Geng et al., describes an interesting finding that CRISPR-Cas9-modified HAP1 cells obtained an unexpected deletion of 10q23.31 region that contains four recurrently deleted genes in human cancers, including the tumor suppressor gene PTEN. The authors initially observed 10q23.31 loss from chromatin profiling data in CRISPR-Cas9-modified HAP1 cells, and further investigated 10q23.31 loss in a panel of diverse human cancer patient cells using public databases.

Overall, this manuscript demonstrated an important recurrent phenomenon caused by CRISPR-Cas9-mediated gene editing in HAP1 cells, which may represent a general caveat of using CRISPR-Cas9-mediated gene editing in other cell lines as well and reflect the similar genetic event that occurs in diverse cancer patient cells. The authors provided some mechanistic and clinical insights on the recurrent 10q23.31 loss in human cancers. The authors have strengthened the manuscript and the data is supportive of the conclusion.

A few minor things: some typos, eg. line 414 (original 700) still not fixed (differ. to); may consider include some of the rebuttal data to supplementary figures to improve reading experience/understanding for particular interested readers. I have no further suggestions.

Referee Cross-Comments:

This reviewer agree with Reviewer#2 that the mechanism underlying the findings in this manuscript is still unclear, which is common that one story can not explain everything. It would be great if the authors can follow up and elucidate more in the future work. In terms of ZnF orTALEN beyond CRISPR, it would be interesting also important to determine whether other different genome editing stress and even numerous other stimuli can also cause similar effects or not. But this maybe a little out of scope here.

We thank the reviewers for their second assessments of our work and have implemented the changes as requested.

Reviewer #1

In their research article, Geng et al. made an unexpected discovery regarding a 287 kb region on Chromosome 10 (10q23.31) in chronic myelogenous leukemia HAP1 cells, commonly used in CRISPR/Cas9 screens. They found a deletion of this region consisting of four protein coding genes: PAPSS2, ATAD1, KLLN, and PTEN. Interestingly, the deletion was not attributed to CRISPR/Cas enzymatic activity but was rather generated or promoted during the CRISPR-Cas engineering process itself. The deletion had significant consequences, as it led to global changes in histone acetylation and gene expression, affecting vital cellular processes like the cell cycle and DNA replication.

The reviewer previously suggested that there is also the possibility that the deletion occurred in a subset of HAP1 cells. The reviewers suggested that the authors should discuss the possibility of clonal evolution or provide evidence for the presence or absence of the clonal deletion in HAP1 cells. Additionally, the authors should investigate the presence of off-target sites during sgRNA design or provide evidence that no homologous sequences are found at the Chr10 deletion boundaries. Finally, it is unclear whether the unmodified clones and dt-dt1 clones are related. Providing information on the junction sequences at the 10q23.31 deletion and their clonality would be beneficial for better understanding the findings.

The reviewer's major and minor comments were fully addressed by the authors. First, the authors discussed the clonal evolution effect in the discussion section. Second, the authors demonstrated that the deletion at Chr10 was not derived from CRISPR/Cas off-target activity. Third, the authors demonstrated that the PAPSS2-PTEN locus was already lost in a considerable number of cells/alleles in the "bulk" HAP1 population (Fig. R1). Lastly, the authors concluded the junction for the dt1 clone with the ChIP-seq reads, and the authors acknowledged that the breakpoint was not defined by split reads, as shown in Figure R2, and concluded that this is a presumed location for the dt1 clone.

Minor comments:

The reviewer suggested that the authors include Figure R1 and Figure R2 in the revised manuscript to support their discussion of clonal evolution, but the decision is up to the authors. Otherwise, the reviewer is generally satisfied with the revised manuscript.

We have moved Figure R1 and Figure R2 and refer to the figures in the revised manuscript. The figures are available under Fig. S1 and Fig. S2. Figure legends were updated accordingly.

Reviewer #2

The response to point 1 concludes that the deletion of the PAPSS2-PTEN locus in HAP1 cells can be induced by exposure to cellular stress (i.e. treatment with different selection antibiotics such as blasticidin, G418 or puromycin), in a dose-dependent manner nonetheless. These are fairly specific cellular stressors, so this should be considered in the messaging in the manuscript, and may provide much more insight into the mechanism behind the PAPSS2-PTEN if contrasted to other cellular stressors.

It was disappointing to see that the authors did not address point 2 and demonstrate that the PAPSS2-PTEN locus deletion could also be induced by non-CRISPR/Cas9 genome editing technologies such as ZnF or TALEN approaches, for which there are many reagents readily available for targeting various human loci. My expectation is that the genome engineering strategy

at play for aggravating the 10q23.31 deletion is probably irrelevant, and the authors have certainly not ruled this out.

The CRISPR system remains the most powerful and widely used genome-editing tool. TALENs/ZFNs genome engineering may be developed further to become more broadly applicable in the future and would justify a separate large-scale study dedicated only to benchmarking and comparing of the different genome engineering methods with regards to prevention of unintended genomic aberrations.

Reviewer #3

The paper by Geng et al., describes an interesting finding that CRISPR-Cas9-modified HAP1 cells obtained an unexpected deletion of 10q23.31 region that contains four recurrently deleted genes in human cancers, including the tumor suppressor gene PTEN. The authors initially observed 10q23.31 loss from chromatin profiling data in CRISPR-Cas9-modified HAP1 cells, and further investigated 10q23.31 loss in a panel of diverse human cancer patient cells using public databases.

Overall, this manuscript demonstrated an important recurrent phenomenon caused by CRISPR-Cas9-mediated gene editing in HAP1 cells, which may represent a general caveat of using CRISPR-Cas9-mediated gene editing in other cell lines as well and reflect the similar genetic event that occurs in diverse cancer patient cells. The authors provided some mechanistic and clinical insights on the recurrent 10q23.31 loss in human cancers. The authors have strengthened the manuscript and the data is supportive of the conclusion.

A few minor things: some typos, eg. line 414 (original 700) still not fixed (differ. to); may consider include some of the rebuttal data to supplementary figures to improve reading experience/understanding for particular interested readers. I have no further suggestions.

We have corrected grammar and spelling errors.

We have also included data from the rebuttal to the supplemental figures. We kindly refer this reviewer to our response of reviewer 1.

Referee Cross-Comments:

This reviewer agree with Reviewer#2 that the mechanism underlying the findings in this manuscript is still unclear, which is common that one story can not explain everything. It would be great if the authors can follow up and elucidate more in the future work. In terms of ZnF orTALEN beyond CRISPR, it would be interesting also important to determine whether other different genome editing stress and even numerous other stimuli can also cause similar effects or not. But this maybe a little out of scope here.

We consider our study as a foundation for future studies that are designed to investigate the underlying mechanism of our findings, as well as safety and specificity towards genomic aberrations upon diverse cell stress conditions.

November 7, 2023

RE: Life Science Alliance Manuscript #LSA-2023-02128-TRR

Dr. Claudia Kutter
Karolinska Institutet
Microbiology, Tumor and Cell Biology, Science for Life Laboratory
Solnavagen 9
Stockholm 171 77
Sweden

Dear Dr. Kutter,

Thank you for submitting your Research Article entitled "Intrinsic deletion at 10q23.31, including the PTEN gene locus, is aggravated upon CRISPR Cas9-mediated genome engineering in HAP1 cells mimicking cancer profiles". It is a pleasure to let you know that your manuscript is now accepted for publication in Life Science Alliance. Congratulations on this interesting work.

DISTRIBUTION OF MATERIALS:

Again, congratulations on a very nice paper. I hope you found the review process to be constructive and are pleased with how the manuscript was handled editorially. We look forward to future exciting submissions from your lab.

Sincerely,
